# Hypoxic-ischemic brain injury in neonatal mice sequentially recruits neutrophils with dichotomous phenotype and function

Mathis Richter [1,2,3,11], Eva Diesterbeck[1,2,11], Ekaterina Pylaeva [4,5], Nicole Labusek[1,2], Christian Köster[1,2], Dennis Nagel [6], Laura Karsch[6], Alexa Josephine Fischer[1,2], Marah Sous[1,2], Marcel Jung [6], Raphael Chevre[3], Nina Hagemann[2,7], Erik Axel Andersson[8,9], C. Joakim Ek[9], Vikramjeet Singh [6], Dirk M. Hermann [2,7], Matthias Gunzer [6,10], Jadwiga Jablonska [4,5], Ursula Felderhoff-Müser[1,2], Ivo Bendix [1,2], Oliver Soehnlein [3] & Josephine Herz[1,2] ✉

Neonatal encephalopathy caused by hypoxia-ischemia (HI) leads to a strong neutrophil infiltration. The long-held assumption that neutrophils act exclusively as tissue-damaging cells, is challenged by increasing evidence of a profound neutrophil heterogeneity. Here, we uncovered a pronounced phenotypical and functional diversification of neutrophils in neonatal mice depending on the disease stage. Neutrophil infiltration was biphasic, peaking 1 and 7 days after HI. Early brain-infiltrating neutrophils displayed a hyperactivated phenotype, whereas neutrophils at day 7 exhibited an angiogenic phenotype with high Siglec-F expression. Acute neutrophil depletion protected against neural cell death, associated with decreased hyperactivity in adolescent animals. Delayed neutrophil depletion impaired vascular and oligodendrocyte regeneration, resulting in exacerbated alterations of anxiety-related behavior and myelination deficits. These findings suggest a divergent function of neutrophils, with early neutrophils aggravating tissue damage and late neutrophils contributing to neurological recovery. The disease stage-dependent neutrophil diversification offers new possibilities to identify disease-stage-specific therapeutic targets.

Neonatal encephalopathy due to hypoxia-ischemia (HI) is a leading cause of death and disability in children worldwide. To date, the only available therapy is hypothermia, which is, however, limited by a short therapeutic window[1]. The identification of novel therapeutic targets in the latent, secondary and tertiary disease phase requires a better understanding of HI-pathophysiology, which involves a pronounced infiltration of peripheral leukocytes, with neutrophils being a major cell population.

Early recruitment of neutrophils into the ischemic brain contributes to acute neuronal degeneration in models of adult brain ischemia and neonatal HI[2–5]. However, the majority of studies focused on the acute disease phase neglecting potential effects on secondary injury and inflammatory processes overlapping with endogenous repair mechanisms[6]. While in adult mice the peak of neutrophil infiltration was observed 1–3 days after the insult followed by a continuous decline[7,8], different kinetics may apply to neonatal HI, as the CNS and the immune system are still developing. Neurodevelopmental processes lasting into early adulthood include for example oligodendrocyte maturation and vascularization[9,10]. Regarding the immune system, neonatal neutrophils are suggested to be immature, with

reduced migration to sites of inflammation and less reactive oxygen species (ROS) production[11]. However, we recently showed that neonatal neutrophils are highly responsive to HI-induced sterile tissue injury, with neutrophils being activated in the injured brain and promoting acute neurodegeneration[3].

The long-held assumption that neutrophils are uniform in function and phenotype, serving exclusively as pro-inflammatory and tissue damaging cells is challenged by increasing evidence of a profound phenotypic and functional neutrophil heterogeneity[12]. Neutrophils may not only induce degeneration, but can also participate in resolution of inflammation and repair, as recently suggested in models of vascular inflammation, cancer, sterile liver inflammation and optic nerve injury[13–18]. In the context of ischemic brain injury, neutrophil diversity is poorly understood. Previous work in adult experimental stroke showed a positive association between neuroprotection and exogenously induced N2 polarization[19], although the causal relationship between the neutrophil switch and tissue protection was missing. Furthermore, it remains unclear whether ischemic brain damage might endogenously induce neuroprotective neutrophils for initiation of repair. While recent transcriptomic profiling of peripheral and brain-infiltrating neutrophils uncovered a neutrophil diversification depending on the disease phase[20], potential functional differences remain unknown.

In the present work, we demonstrate an unexpected phenotypical and functional neutrophil dichotomy following neonatal HI with brain-infiltrating neutrophils contributing to tissue damage in the acute disease phase while promoting neurological recovery in the delayed disease stage. This functional shift is associated with phenotypical changes, revealed by overt ROS production of early-infiltrating cells while late-infiltrating neutrophils display a pro-angiogenic phenotype associated with the appearance of a Siglec-F[high] neutrophil population that can be induced by GM-CSF stimulation in vitro.

## Results

### Biphasic neutrophil infiltration into HI-injured brain tissues is associated with changes in the molecular tissue environment and different neutrophil migration responses

Neutrophil infiltration kinetics were characterized by flow cytometry and light sheet microscopy until 10 days after neonatal HI. Flow cytometry analyses demonstrated a peak of neutrophil infiltration 1 and 7 days after the insult, while neutrophil numbers were similar between sham and HI animals at 12 h, 3 d and 10 d after HI (Fig. 1A, Supplementary Fig. S1A). The biphasic neutrophil recruitment was confirmed by three-dimensional (3D) whole brain imaging in cleared brain tissues of Catchup[IVM] mice with tdTomato-expressing neutrophils[21], revealing an intraparenchymal tissue infiltration (Fig. 1B, Supplementary Movie S1). Whether neutrophil infiltration at the different disease stages is associated with tissue injury was assessed via immunohistochemistry, revealing a negative correlation between infiltrated neutrophils and neuronal densities at day 1, while no clear association was observed at day 7 after HI (Supplementary Fig. S1B). Taking different neutrophil infiltration routes into account[22–24], two-photon imaging was performed in brain tissues with intact skull, showing neutrophils in the subcortical meninges and skull bone marrow (Fig. 1C dorsal view, Supplementary Fig. S1C). However, a significant intraparenchymal tissue infiltration was also confirmed (Fig. 1C). Detailed regional distribution was assessed via immunohistochemistry, revealing the largest proportion of neutrophils in the thalamus, followed by the meninges, hippocampus and cortex at day 1 after HI (Fig. 1D, Supplementary Fig. S1D). Interestingly, 7 days after HI the proportion of neutrophils increased in the hippocampus and declined in the thalamus and meninges (Fig. 1D, Supplementary Fig. S1D). Localization with regard to vessel distances was analyzed in 3 mm thick brain tissue sections via confocal microscopy (Fig. 1E), revealing a closer association with the vasculature 7 days after HI (Fig. 1E). Confocal microscopy in 20 μm tissue sections stained for

CD31, Laminin and anti-Ly6G enabled more detailed analyses, distinguishing intravascular, perivascular, transmigrating and clearly intraparenchymal neutrophils (Fig. 1F, Supplementary Fig. S1E). At 1 day after HI, more than half of neutrophils were located in the vasculature, while up to 60% were localized in the parenchyma and up to 20%, either within or crossing the barrier of the perivascular space 7 days after HI (Fig. 1F, Supplementary Fig. 1E). Confirming results from 3D analyses (Fig. 1E), we detected smaller vessel distances of intraparenchymal neutrophils at day 7 (Fig. 1F).

To identify neutrophil-attracting signals in the brain, we quantified expression of a broad set of proteins involved in tissue inflammation, protection and repair with proteome profiler antibody arrays (Fig. 2A). To correct for physiological changes due to development in sham-operated animals (Supplementary Fig. S2A), values from HI-injured mice were normalized to mean values of sham mice from the same analysis time point. Focusing on candidates with significant results in ANOVA tests and mean fold change values above 2 or below 0.5, we identified 30 proteins out of 111 candidates with a differential expression (Fig. 2A, Supplementary Fig. S2B). One class of proteins showed a particular upregulation at 24 h and 7 d, i.e. osteopontin, myeloperoxidase, Chitinase 3-like 1 and CCL6, with osteopontin revealing the most consistent and pronounced increases at both time points (Fig. 2A, Supplementary Fig. S2B). However, we also identified a number of proteins, that were specifically upregulated 7 days after the insult, including chemokines, e.g. CXCL-2, CXCL-10 and CCL5 (Fig. 2A, Supplementary Fig. S2B). Furthermore, a subset of proteins was not detectable either at day 1 or day 3, but induced at other time points. These included CXCL-1, which was upregulated at day 1 but downregulated at day 7, while CXCL-5, Lipocalin-2 and MMP-9 were selectively induced at day 7 after HI (Fig. 2B, Supplementary Fig. 2B). In addition to the unique regulation of classical neutrophil chemoattractants intrinsic differences in neutrophils may also contribute to the dynamic changes in neutrophil infiltration. Interestingly, CD11b expression on brain neutrophils resembled neutrophil infiltration dynamics with peak expression levels 1 and 7 days after HI, showing a positive correlation with the number of infiltrated neutrophils (Fig. 2C, Supplementary Fig. 2C). To characterize potential developmental differences in neutrophil migration responses, we screened their migration capacities to a variety of classical neutrophil chemoattractant signals and selected proteins, based on their expression pattern in the brain (Fig. 2A, B) in ComplexEye assays[25] (Fig. 2D, Supplementary Fig. S3). Compared to vehicle controls, significant responses were observed for typical neutrophil chemoattractants, i.e. CXCL1, CXCL2, CXCL5, LTB4 and GM-CSF (Fig. 2D, E, Supplementary Fig. S3). Notably, stimulation responses were independent of HI but depended on the developmental stage. For instance, migration speed in response to CXCL1 and CXCL2 were increased at 7 days compared to earlier time points, while the response to GM-CSF was higher at day 1 (Fig. 2D, E). In addition to chemokines, we detected a unique upregulation of several cytokines (e.g. interleukin (IL)-1alpha, IL-13 and IL-33) and proteins with neuroprotective and regenerative functions, e.g. Gas 6, LIF, Pentraxin 3 and Adiponectin in the brain 7 days after HI (Fig. 2A, Supplementary Fig. S2B, Supplementary Table S1). Interestingly, among these proteins, the angiogenesis-promoting proteins Chemerin and Angiopoietin were also elevated (Fig. 2A, Supplementary Fig. S2B, Supplementary Table S1). VEGF, which was significantly reduced at day 1, recovered 7 days after HI (Fig. 2A, Supplementary Fig. S2B). Together, these results show that the HI-injured brain undergoes pronounced changes during the time course of disease, which may have switched neutrophils' phenotype from a pro-inflammatory and degenerative phenotype in the acute disease phase to a protective and pro-regenerative cell type at the later disease stage.

### Functional dichotomy of early vs. late infiltrating neutrophils

To assess the functional implication of neutrophils at the two time points, we depleted them either at 24 h or at 7 d after HI and evaluated

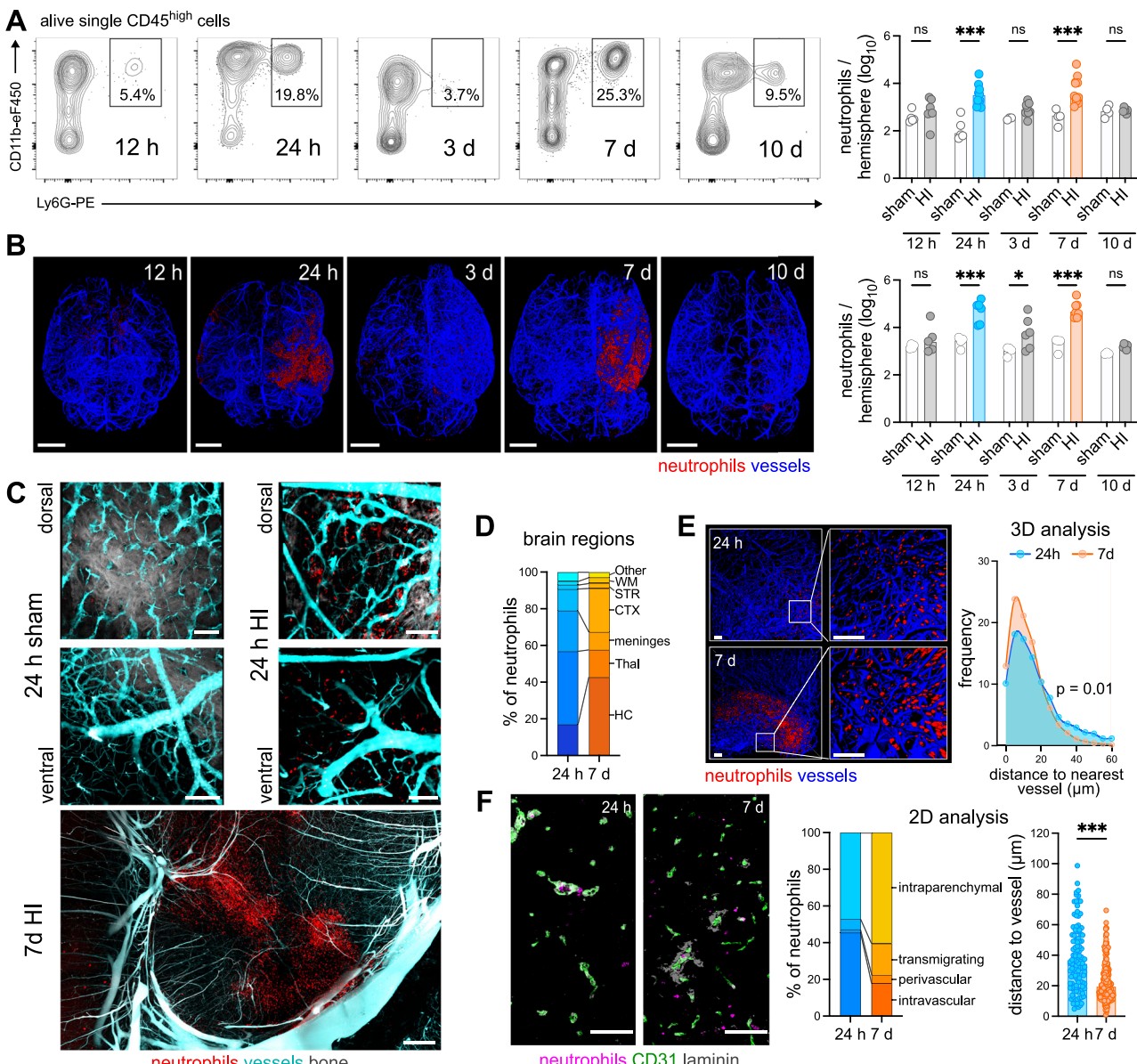

**Fig. 1 | Biphasic cerebral neutrophil infiltration into HI brains. A** Neutrophil cell counts were analyzed in ipsilateral hemispheres. Left: Representative contour plots of CD11b+Ly6G+ neutrophils from pre-gated single alive leukocytes (Supplementary Fig. S1A). $n = 5$ (sham 12 h/ 24 h), $n = 3$ (sham 3 d), $n = 4$ (sham 7 d/ 10 d), $n = 7$ (HI 12 h), $n = 10$ (HI 24 h), $n = 8$ (HI 3 d), $n = 11$ (HI 7 d), $n = 6$ (HI 10 d); two-way ANOVA followed by Šídák's test. **B** Analyses of neutrophil infiltration in ipsilateral (right) hemispheres via light sheet microscopy in Cubic-cleared brain tissues. Vessel structures (blue) were visualized by FITC-gelatin perfusion in Catchup[IVM] mice with tdTomato expressing neutrophils (red), Scale bar: 2 mm, $n = 4$ (sham 12 h/ 24 h/ 3 d/ 7 d), $n = 3$ (sham 10 d), $n = 5$ (HI 12 h), $n = 6$ (HI 24 h/ 3 d/ 10 d), $n = 7$ (HI 7 d), two-way ANOVA followed by Šídák's test. **C** Exemplary two-photon images of neutrophils (red) in dorsal-ventral direction revealing sub-cortical associated neutrophils (top) and intraparenchymal neutrophils (middle) 24 h and 7 d after sham operation or HI. Images are representative for 1 sham (24 h), 2 HI (24 h) and 1 HI (7 d) mouse. Scale

bar: 100 μm (24 h), 200 μm (7 d) **D** Quantification of neutrophil localization in different brain regions. $n = 6$ (24 h), $n = 8$ (7 d) **E** Quantification of neutrophil distribution related to the vasculature in the hippocampal region of 3 mm thick cleared tissue sections via confocal microscopy. The distance of neutrophils from $n = 5$ (24 h) and $n = 7$ (7 d) mice was quantified. Distribution of the distances was compared using the two-tailed Kolmogorov-Smirnov test. Scale bar: 100 μm **F** Analyses of neutrophil localization in relation to the vasculature via immunohistochemistry (according to Supplementary Fig. S1E), two-tailed Mann Whitney test. $n = 6$ (24 h), $n = 8$ (7 d). Scale bar: 50 μm. Bar graphs show median values in A/B and means in D/F. Individual data for D/F are presented in Supplementary Fig. S1D, E and the source data file. n.s.: not significant, $*p > 0.05$, $***p < 0.001$. Exact $p$ values are given in the source data file. HC hippocampus, Thal thalamus, STR striatum, CTX cortex, WM white matter.

long-term neurobehavioral deficits in both treatment groups (Fig. 3A). Using an improved neutrophil depletion protocol combing anti-Ly6G with anti-rat IgG[26], we obtained a significant reduction of both, circulating and brain-infiltrating neutrophils after early and late neutrophil depletion, while other leukocyte subsets were not affected in either both treatment protocols (Supplementary Fig. S4A, B); and a similar recovery of neutrophil numbers was obtained 5 days after the last antibody injection (Supplementary Fig. S4C). Recovered neutrophils

revealed a slightly more immature phenotype, i.e. reduced expression of CD101, CXCR2 and Ly6G, while a broad set of other markers was not affected (Supplementary Fig. S4D). Physiological weight gain in sham-operated animals was not altered by induced neutropenia (Supplementary Fig. S5A). However, in line with previous reports[27,28], HI induces an acute decrease in weight gain, which was similar in all treatment groups (Fig. 3B). Percent weight gain between 0-3 days and 7-14 days after HI was similar regardless of treatment and time point

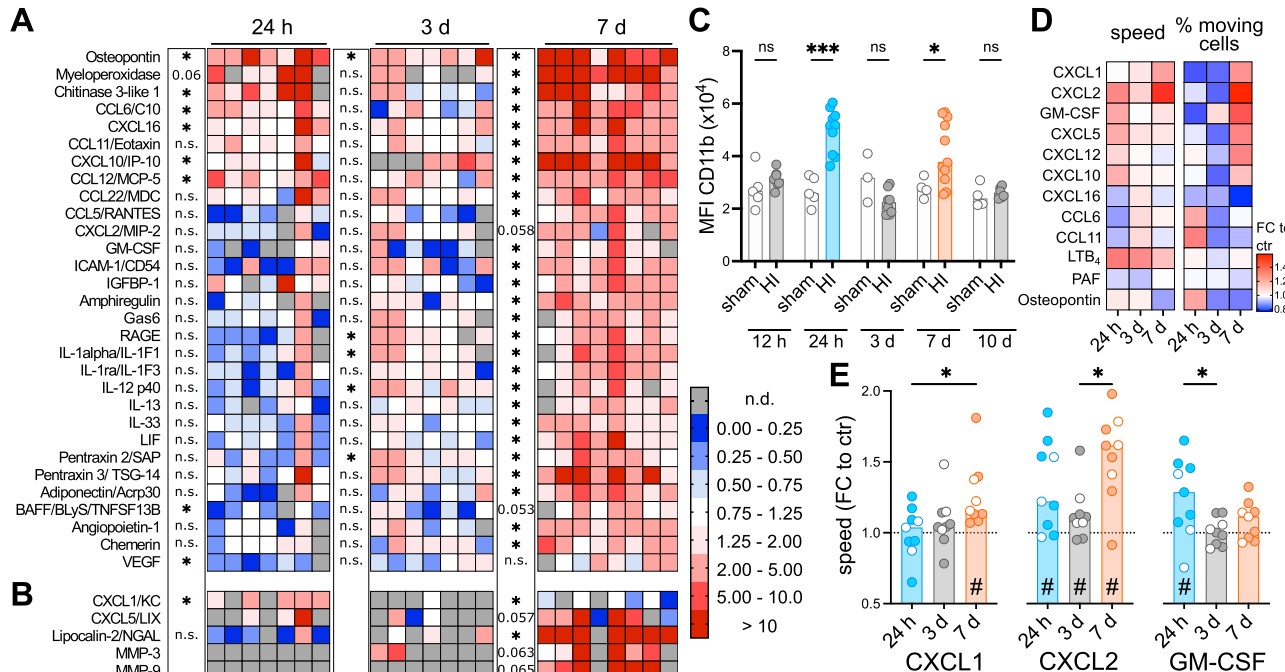

**Fig. 2 | Disease-stage-dependent protein expression levels in brain tissues and blood neutrophil migration patterns. A** Protein expression analyzed with Proteome Profiler antibody arrays in ipsilateral brain tissues at the indicated time points after HI. Fold change values compared to sham animals of the same time point. Differentially expressed genes were determined based on ANOVA testing and fold change values < 0.5 and >2.0, details are provided in the source data file, $n = 4$ (sham 24 h/ 3 d/ 7 d), $n = 7$ (HI 24 h/ 3 d), $n = 8$ (HI 7 d). Results of two-tailed one-sample t-test or Wilcoxon signed rank test are shown for each time point. Individual data and results for comparisons between time points are presented in Supplementary Fig. S2B and the source data file. **B** Proteins, for which ANOVA-tests could not be applied due to signal intensities below detection limit at one of the 3 assessed time points, but revealing fold change values below 0.5 and above 2.0 are shown to visualize induction of protein expression at certain time points. Details are provided in the source data file. **C** CD11b expression on brain-infiltrating

neutrophils assessed via flow cytometry $n = 5$ (sham 12 h/ 24 h), $n = 3$ (sham 3 d), $n = 4$ (sham 7 d/ 10 d), $n = 7$ (HI 12 h), $n = 10$ (HI 24 h), $n = 8$ (HI 3 d), $n = 11$ (HI 7 d), $n = 6$ (HI 10 d), two-way ANOVA followed by Šídák's test. **D** Characterization of blood neutrophil migration patterns via ComplexEye measurements in response to different stimuli. Mean fold-change values to vehicle controls form $n = 3$ sham and $n = 6$ HI mice/time point. Individual data and results for comparisons between time points are presented in Supplementary Fig. S3 and the source data file. **E** Normalized mean migration speed of neutrophils in response to CXCL1, CXCL2 and GM-CSF. $n = 9$ per time point ($n = 3$ sham (white circles), $n = 6$ HI (filled circles). *One-way ANOVA followed by Šídák's (CXCL1/2), Kruskal-Wallis followed by Dunn's (GM-CSF). # tow-tailed one-sample t test (CXCL1/2) or Wilcoxon signed rank (GM-CSF). Bar graphs in C/E show median values. n.s.: not-significant, *$p < 0.05$, ***$p < 0.001$. Individual values and exact $p$ values are given in the source data file.

(Supplementary Fig. S5B), excluding potential confounding effects on neurobehavioral testing by differences in physical development. Long-term neurodevelopment was assessed in the open field (OF) and elevated plus maze (EPM) in adolescent animals 5 weeks after neonatal HI to determine activity and anxiety-related behavior (Fig. 3A). As a sign of hyperactivity, we observed increased movement velocities and mobility in HI-injured animals in the OF (Fig. 3C). HI-induced alterations in anxiety-related behavior were demonstrated by more movement in the open arms of the EPM (Fig. 3D). Neither early nor late neutrophil depletion affected behavioral development in sham-operated animals (Supplementary Fig. S5C, D). However, in HI-injured mice early depletion prevented the development of hyperactivity (Fig. 3C). Interestingly, while late depletion did not affect this neurodevelopmental deficit, we detected a significant deterioration of HI-induced alterations in anxiety, demonstrated by an increased time in the open arms of the EPM (Fig. 3D). These results suggest that neutrophils in the acute disease phase contribute to development of hyperactivity, while changes in anxiety-related behavior are attenuated by secondarily infiltrating neutrophils.

To assess whether these neurodevelopmental changes were associated with long-term tissue injury, we quantified tissue atrophy in cresyl-violet stained tissue sections, showing a less pronounced HI-induced long-term tissue loss in early-depleted mice, while in late-depleted animals an overall aggravated tissue injury was observed (Fig. 3E). These results are consistent with long-term alteration of HI-

induced functional deficits, as demonstrated by a positive correlation with HI-induced alteration of anxiety-related behavior and hyperactivity, observed in the EPM and OF, respectively (Fig. 3F). Quantification of mRNA levels for neuronal (NeuN, MAP2) and myelination (CC1, CPNase, MAG, MBP) proteins largely confirmed these regulations, demonstrated by a reduced expression in isotype-treated HI mice, which was partially improved by early neutrophil depletion, while late depletion resulted in more severe reductions compared to isotype controls (Fig. 3G, Supplementary Fig. S5E). Similarly, MBP protein expression was more severely decreased in mice with late neutrophil depletion compared to early anti-Ly6G or isotype control treatment (Fig. 3H, Supplementary Fig. S6).

**Early brain-infiltrating neutrophils induce neurodegeneration associated with myeloid cell accumulation, while late-infiltrating neutrophils promote oligodendrocyte regeneration**

Potentially different cellular targets of early *vs.* late brain-infiltrating neutrophils were analyzed by immunohistochemistry 10 days after HI (Fig. 4A), when secondary injury and inflammatory processes overlap with the initiation of regenerative responses[6]. The previously reported detrimental effect of early neutrophils[3] is verified in the present study, as demonstrated by an increased neuronal density (Fig. 4B) after early neutrophil depletion, while a slightly more severe neuronal loss was observed in the absence of neutrophils at the delayed disease stage (Fig. 4B). To determine whether this was associated with differences in

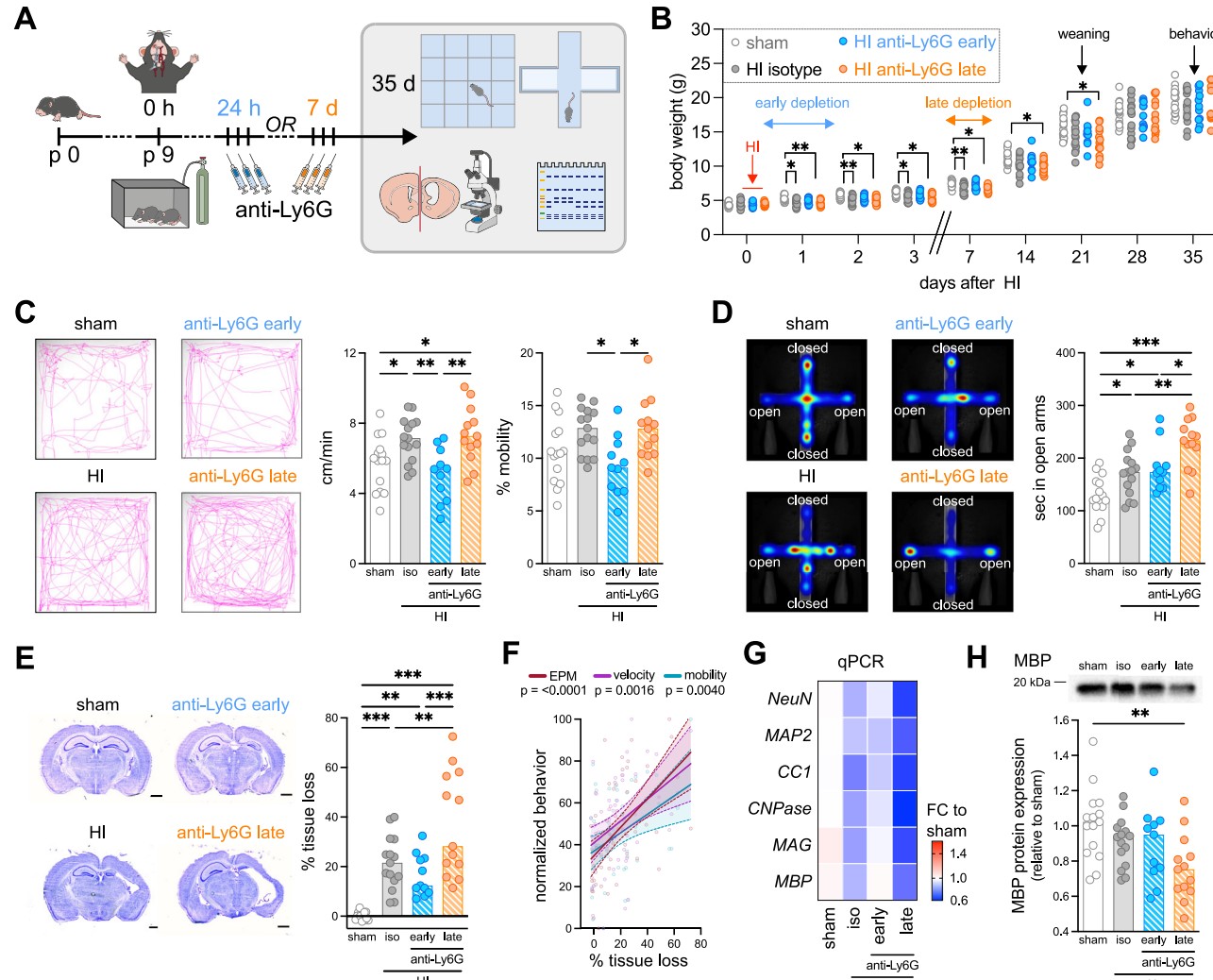

**Fig. 3 | Early and late neutrophil depletion result in opposite neurodevelopmental outcome. A** Schematic experimental design showing analysis timeline and readouts to assess the impact of neutrophil depletion on HI-induced long-term neurodevelopmental deficits. Neutrophils were depleted by combing anti-Ly6G with anti-rat IgG treatment at 12 h/36 h/50 h (early depletion) or 6/7/8 days (late depletion) after HI. **B** Body weight development over the time course of the experiment, two-way-repeated measurement ANOVA followed by Tukey's multiple comparisons test. **C** HI-induced hyperactivity-related behavior analyzed in the Open Field test by quantification of mean velocities and mobility. Representative images reveal movement tracks over the observation period of 5 min. **D** Anxiety-related behavior was evaluated in the Elevated Plus Maze by measurement of the time spent in the open arms of the maze. Representative images reveal heat maps summarizing localization of mice over the observation period of 5 min. **E** Cresyl staining and quantification of total hemisphere atrophy 6 weeks after HI Scale bar:

1 mm. **F** Correlation of normalized behavioral parameters (lowest value: 0, highest value: 100) from (**C**) and (**D**) with tissue atrophy from (**E**), simple linear regression, linear regression lines with 95% confidential intervals are shown for each readout. **G** qPCR of different neuronal (*NeuN, MAP2*) and myelination (*CC1, CPNase, MAG, MBP*) transcripts. Mean fold change values compared to sham isotype treatment are represented. Individual data and results for comparisons between experimental groups are presented in Supplementary Fig. S5E and the source data file. **H** Western blot analysis of MBP protein expression, normalized to the total protein. Uncropped and unprocessed images of gels and blots are provided in Supplementary Fig. S6. Bar graphs in C/D/E/H show median values, one-way ANOVA followed by Holm-Šídák's multiple comparisons test. *n* = 8 sham isotype (4 early, 4 late), *n* = 8 sham anti-Ly6G (4 early, 3 late), *n* = 15 HI isotype (8 early, 7 late), *n* = 11 HI anti-Ly6G early, *n* = 13 HI anti-Ly6G late. *$p < 0.05$, **$p < 0.01$, ***$p < 0.001$. Individual values and exact *p* values are given in the source data file.

secondary apoptosis or neurogenesis, we quantified the number of TUNEL positive cells, which was elevated in all HI-injured animals with no modulation by anti-Ly6G treatment (Supplementary Fig. S7A). As a measure of neurogenesis, we quantified the number of doublecortin (DCX) positive cells in the hippocampus and of proliferating cells in the subgranular zone (SGZ), both being increased in HI-injured animals, independent of early or late neutrophil depletion (Supplementary Fig. S7B, C).

With regard to HI-induced glial activation, we assessed GFAP and Iba-1 immunoreactivity, as markers for astrogliosis and myeloid/microglia cell accumulation. Both were reduced after early neutrophil depletion, while animals with late depletion were comparable to isotype-control HI mice (Fig. 4C, Supplementary Fig. 7D). Since Iba-1

cannot discriminate between different myeloid cell subsets, we performed spectral flow cytometry for a broad set of cell surface markers to distinguish different macrophage/microglia subtypes (Fig. 4D, Supplementary Fig. S7E). While HI led to the appearance of different disease-associated macrophage subsets in the ipsilateral hemisphere 10 days after HI, neither early nor late neutrophil depletion resulted in marked changes of myeloid cell subsets 10 days after HI (Fig. 4D, Supplementary Fig. S7E). HI-induced oligodendrocyte loss was prevented by early neutrophil depletion, while the absence of neutrophils in the later stage of disease led to a further reduction of oligodendrocytes (Fig. 4E). As an endogenous response for recovery, oligodendrocytes proliferate. Interestingly, HI-induced compensatory oligodendrocyte proliferation was decreased after late neutrophil

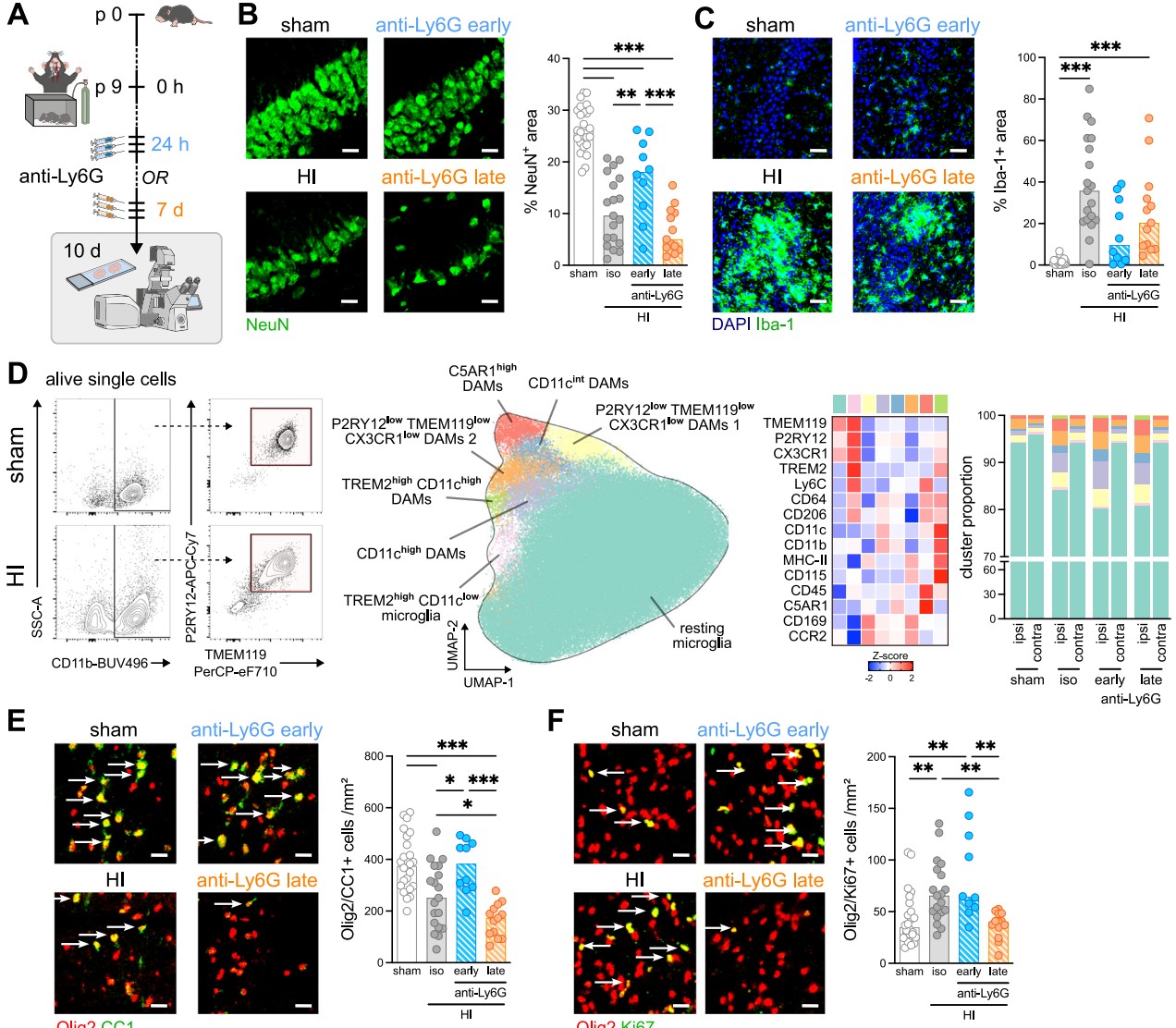

**Fig. 4 | Dichotomous impact of early and late brain-infiltrating neutrophils on neonatal HI-induced brain injury and repair. A** Schematic experimental design showing analysis time line and readouts to determine the effect of early and late neutrophil depletion on brain tissue damage and regeneration. **B** Neuronal density in the hippocampus was quantified via immunohistochemistry for the pan-neuronal marker NeuN. One-way ANOVA followed by Holm-Šídák's multiple comparisons test. **C** Myeloid cell accumulation was quantified via immunohistochemistry for Iba−1. Kruskal-Wallis followed by Dunn's multiple comparison test. **D** Macrophage/microglia cell composition analyzed via spectral flow cytometry. Gating strategy (left and Supplementary Fig. 7), UMAP dimensional-reduction with FlowSOM clustering (middle) and heatmap displaying z-scores for surface marker expression in different clusters including quantification of the relative abundances

of identified cell populations in ipsilateral or contralateral hemispheres (right). $n = 5$ sham (3 isotype (1 early, 2 late), 2 anti-Ly6G (1 early, 1 late), $n = 7$ HI isotype (3 early, 4 late), $n = 6$ HI anti Ly6G early, $n = 6$ HI anti-Ly6G late. Individual data are presented in Supplementary Fig. S7E . Mature (**E**) and proliferating (**F**) oligodendrocytes were quantified in the white matter via analyses of Olig2(red)/CC1 (green) (**E**) and Olig2(red)/Ki67(green) (**F**) double positive cells (indicated by arrows). One-way ANOVA followed by Holm-Šídák's multiple comparisons test in **E** and Kruskal-Wallis followed by Dunn's test in **F**. **B/C/E/F** Scale bars: 50 μm. Bar graphs show median values, $n = 12$ sham isotype (4 early, 8 late), $n = 12$ sham anti-Ly6G (4 early, 8 late), $n = 19$ HI isotype (8 early, 11 late), $n = 10$ anti-Ly6G early, $n = 13$ HI anti-Ly6G late. *$p < 0.05$, **$p < 0.01$, ***$p < 0.001$. Individual values and exact $p$ values are given in the source data file.

depletion (Fig. 4F). Neutrophil depletion did not affect any of the investigated cell populations in sham-operated animals (Supp. Figure S7F). Together with long-term functional outcome data (Fig. 3), these results suggest a detrimental role of early brain-infiltrating neutrophils inducing acute neuronal and oligodendrocyte loss and glial activation, while late-infiltrating neutrophils support recovery by promoting oligodendrocyte regeneration.

## Late neutrophils contribute to vascular remodeling

In addition to the generation of new oligodendrocytes, angiogenesis is a key reparative process following ischemic brain injury[29]. Neutrophils have been implicated in vascular remodeling in various adult disease

entities[16,30]. To characterize developmental vascularization, covering the period of experimental interventions, i.e. between postnatal day 9 (P9, HI induction) until P16 (7 days after HI), we applied light sheet microscopy in large tissue volumes of cleared brain tissues (Fig. 5A, B, Supplementary Fig. S8A–C) followed by automated quantification according to our previously established workflow[31,32]. Total vessel length density continuously increased with development, which was disturbed by neonatal HI at P9 (Fig. 5A, B, Supplementary Fig. S8A–C). Vessel branching particularly increased between P12 and P16 in sham mice, but not in HI-injured mice (Supplementary Fig. S8B). To gain deeper insight into the dynamics of small capillary and larger caliber vessel development, we quantified vessel length densities of vessels

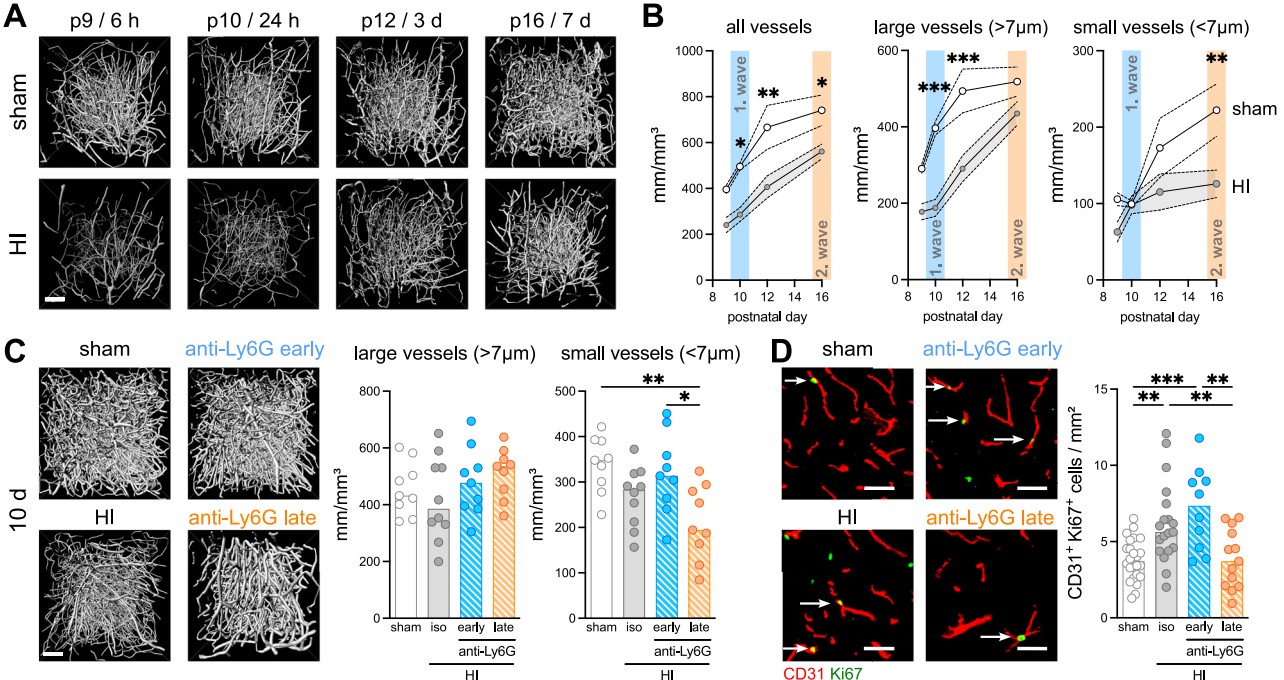

**Fig. 5 | Impact of neonatal HI and neutrophils on developmental vascularization. A** Representative images of 3D-projections of the hippocampal vasculature (306 × 306 x 900 μm) acquired via light sheet microscopy between postnatal day 9 and 16 (P9-P16) in the ipsilateral hemisphere of sham and HI mice. Scale bar: 100 μm. These images were used for quantitative for analyses using VesselExpress[31,32] shown in (**B**, **C**). **B** Quantification of vessel length density of all vessels (left), large vessels (> 7 μm, middle) or small vessels (< 7 μm, right). Mean and SEM values are shown from $n = 4$ sham/time point, $n = 6$ (HI P9), $n = 8$ (HI P10/ P12), $n = 9$ (HI P16), two-way ANOVA followed by Šídák's test. Individual data are shown in Supplementary Fig. S8C and the source data file. **C** 3D projections and quantification of vessel length density 10 days after HI. $n = 4$ sham isotype (2 early, 2 late), $n = 5$ sham anti-Ly6G (2 early, 3 late), $n = 10$ HI isotype (5 early, 5 late), $n = 9$ HI anti-Ly6G early, $n = 9$ anti-Ly6G late, one-way ANOVA followed by Holm-Šídák's multiple comparison test. **D** Endothelial cell proliferation was assessed through analyses of tissue sections stained for CD31 (red) and Ki67 (green). Arrows indicate double positive cells. One-way ANOVA followed by Holm-Šídák's multiple comparison test. Scale bar in all images: 100 μm. Bar graphs in C/D show median values. $n = 12$ sham isotype (4 early, 8 late), $n = 12$ sham anti-Ly6G (4 early, 8 late), $n = 19$ HI isotype (8 early, 11 late), $n = 10$ anti-Ly6G early, $n = 13$ HI anti-Ly6G late. $*p < 0.05$, $**p < 0.01$, $***p < 0.001$. Individual values and exact $p$ values are given in the source data file.

smaller and larger than 7 μm, based on Wälchli et al.[33]. Most of the large vessels are established between P9 and P12, while small capillaries develop at later stages, i.e. from P12 onwards (Fig. 5B, Supplementary Fig. S8C). Neonatal HI induced a strong reduction, particularly of larger vessels during the acute disease phase, which, however, recover until day 7 (Fig. 5B, Supplementary Fig. S8A, C). In contrast, development of small vessels remains impaired, demonstrated by a significantly reduced vessel length density 7 days after HI (Fig. 5B, Supplementary Fig. S7C). Delayed capillary regrowth might be related to an early impact on tip cell numbers, important for sprouting angiogenesis and capillary network formation during development[9]. Quantification of Tip cell numbers showed a unique developmental regulation in sham mice, with a considerable number of cells being present between P9 and P10, but rapidly declining from P12 until P16, when no cells could be detected anymore (Supplementary Fig. S8D). Neonatal HI induces an acute Tip cell loss within the first 24 hours after insult, which recovers until 3 days after HI (Supplementary Fig. S8D).

The impact of neutrophils on HI-induced vascularization deficits was investigated 10 days after HI. (Fig. 5C, Supplementary Fig. S8E). Compared to day 7, small vessel development slightly recovered in isotype-treated HI mice, which was, however, impaired in the absence of late-infiltrating neutrophils (Fig. 5C). Since Tip cells recovered by 3 days after HI (Supplementary Fig. S8D) these cells seem rather unlikely to be targeted by late-infiltrating neutrophils. However, in addition to vascular sprouting from Tip cells, a major mechanism of post-ischemic vascular remodeling is the generation of new endothelial cells. Analyses of endothelial cell proliferation via immunohistochemistry demonstrated an endogenous increase in Ki67[+]

endothelial cells, which was absent in animals with late neutrophil depletion (Fig. 5D). Together, these data suggest that while late-infiltrating neutrophils promote endogenous vascular remodeling and angiogenesis in the sub-acute disease phase, thereby supporting post-ischemic tissue recovery.

## Neonatal HI induces alterations in neutrophil function and heterogeneity depending on the time point of tissue infiltration

To characterize early- and late brain-infiltrating neutrophils regarding their phenotype and function, we performed conventional and spectral flow cytometry analyses and neutrophil organoid co-culture assays. Late-infiltrating neutrophils showed a significant reduction of basal ROS production (Fig. 6A). In accordance with the potential involvement of late neutrophils in angiogenesis (Fig. 5C, D), we further quantified the frequency of VEGFR1[+]CD49[+] neutrophils (Fig. 6B, Supplementary Fig. S9A), previously described as pro-angiogenic neutrophils[34,35]. Compared to early brain-infiltrating neutrophils, late-infiltrating cells contained a larger proportion of this neutrophil subset (Fig. 6B). Pro-angiogenic properties of late infiltrating neutrophils were analyzed in co-cultures with aortic rings, demonstrating a significantly increased vascular sprout length in rings incubated with neutrophils from day 7 compared to rings cultivated with brain-infiltrating neutrophils from day 1 (Fig. 6C). The regenerative response of late brain-infiltrating neutrophils was confirmed in scratch assays with brain endothelial cells. While co-culture with brain neutrophils from day 1 did not modulate spontaneous wound healing (i.e. endothelial migration), 7d-neutrophils led to an increase by 50% compared to untreated cells (Fig. 6D).

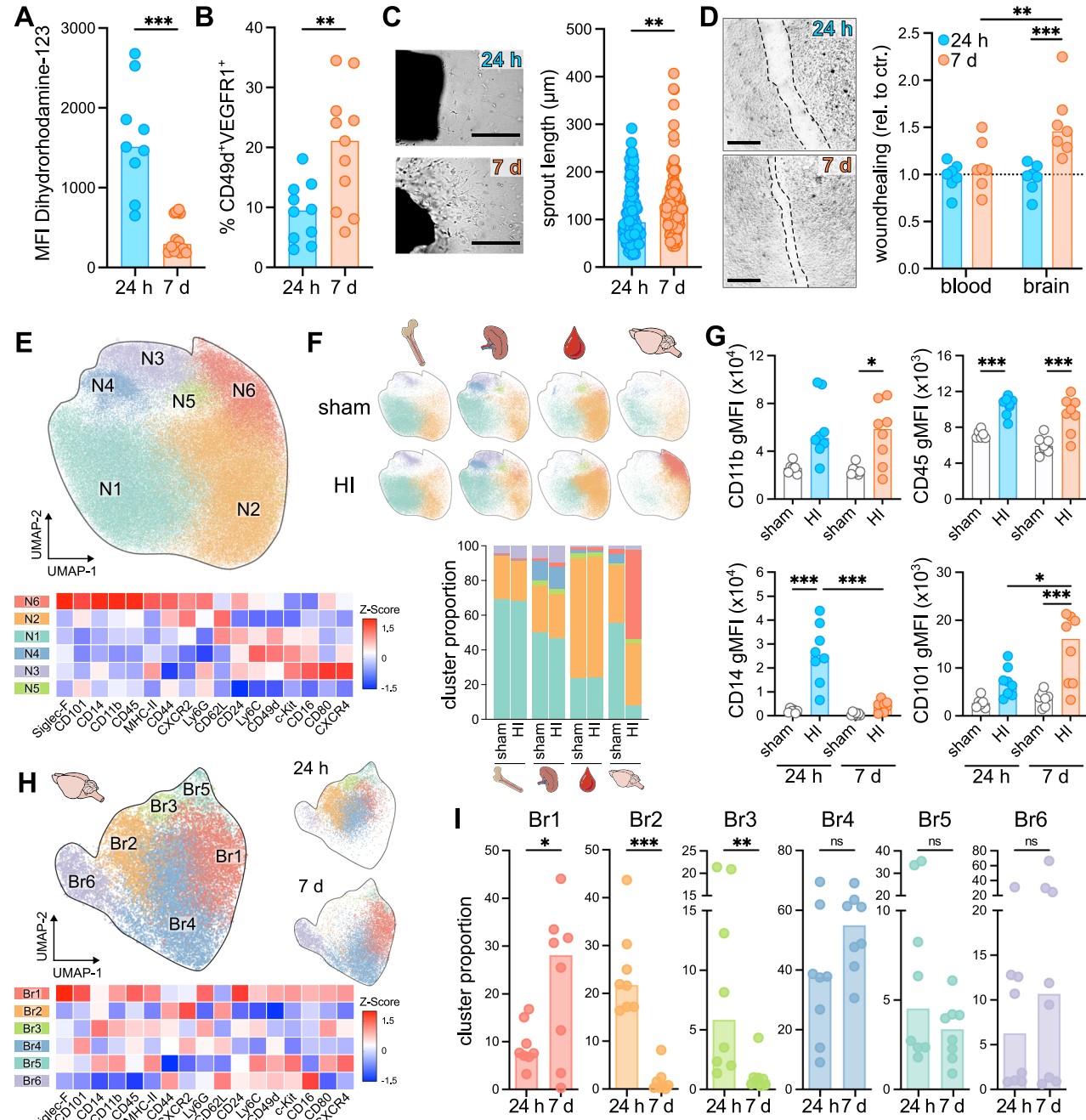

**Fig. 6 | Early and late brain-infiltrating neutrophils differ in phenotype and function. A** Reactive oxygen species production was analyzed by flow cytometry of Dihydrorhodamine-123 (DHR) stained brain neutrophils. Mean fluorescence intensities (MFI) were quantified on FvD⁻, CD45^high, CD11b⁺, Ly6G⁺ neu-trophils (Supplementary Fig. S9A), two-tailed Mann-Whitney test, $n = 9$ (24 h), $n = 14$ (7 d). **B** CD49d⁺VEGFR1⁺ neutrophils were quantified in brain-infiltrating neu-trophils; two-tailed unpaired t-test, $n = 10$ (24 h), $n = 11$ (7 d). **C** Vascular sprout lengths were quantified in aortic rings co-cultured with brain-infiltrating neu-trophils. Three aortic rings/time point, each co-incubated with 20,000 neutrophils from in total $n = 10$ (24 h) and $n = 8$ (7 d) mice, analyzed in two independent experiments, details are provided in the source data file, two-tailed Mann-Whitney test comparing sprout lengths from all 3 rings/ time point, scale bar: 400 μm. **D** Wound healing assessed in scratch assays with brain endothelial cells (bEnd.3) in co-culture with neutrophils isolated from blood or brain. Relative wound healing compared to untreated controls, two-way ANOVA followed by Tukey's test. $n = 7$

(24 h) and $n = 6$ (7 d) biological replicates, i.e. wells, each with 10,000 neutrophils from 1-6 mice / well. Data are derived from 5 independent experiments, i.e. cultures. Details are provided in the source data file. **E, F** Neutrophils from different organs were analyzed by spectral flow cytometry followed by UMAP dimensional-reduction and FlowSOM clustering. Mean expression of surface markers of the six identified clusters (N1-6) and their relative abundances in the different organs are displayed, $n = 6$ (sham 24 h), $n = 7$ (sham 7 d), $n = 8$ (HI 24 h & 7 d). **G** MFIs of CD11b, CD45, CD14 and CD101 were quantified on the total neutrophil population, Kruskal-Wallis followed by Dunn's test (CD11b) or one-way ANOVA followed by Šídák's test (CD45, CD14 & CD101). **H** Re-clustering of neutrophils from HI-injured brains, z-scores for mean surface marker expression in the six identified clusters (Br1-6). **I** Relative abundances of brain clusters at 24 h or 7 d after HI, two-tailed Mann-Whitney test (Br1, Br2, Br5 & Br6) or two-tailed unpaired t-test (Br3 & Br4)). Bar graphs show median values in A-D/G/I. *$p < 0.05$, **$p < 0.01$, ***$p < 0.001$. Individual values and exact $p$-values are given in the source data file.

While these analyses were performed on the bulk neutrophil population, we next conducted spectral flow cytometry to identify subpopulations of neutrophils in CD45+, CD11b+, Ly6Cint, Ly6G+ cells (Supplementary Fig. S9B, Fig. 6E). Bone marrow, spleen and blood samples were also included to assess whether potential differences in brain neutrophils were related to alterations in the periphery (Fig. 6E, F, Supplementary Fig. S9C). Dimensional reduction using the Uniform Manifold Approximation and Projection (UMAP) algorithm followed by unsupervised FlowSOM[36] clustering resulted in six main clusters different in maturation and activation phenotype (Fig. 6E). The most prominent cluster in the bone marrow and spleen (N1) revealed an immature phenotype (CD101low CXCR2int CD49dhigh CD62Lhigh CD24high c-Kitint CD16int), while the most abundant cluster in the blood (N2) showed a more mature phenotype (CD101int CXCR2high CD49dlow CD62Lhigh CD24low c-Kitlow CD16low) (Fig. 6E, F). In addition to N1, the spleen contained another immature cluster (N4), not present in bone marrow and blood (Fig. 6E, F). While the proportion of identified neutrophil clusters was similar in sham and HI-injured animals in all analyzed peripheral organs (Fig. 6F, Supplementary Fig. S9C), re-clustering per organ revealed developmental changes demonstrated by an increase in CD101high clusters (BM1 and BL1) in the bone marrow and blood 7 days after HI (Supplementary Fig. S9D–F). This development continued into adulthood, i.e. CD101 expression on neutrophils increased continuously and CD14 expression declined further until 12 weeks of age (Supplementary Fig. S9G).

In contrast to remote organs, neutrophils in the brain acquired a disease-specific phenotype with the emergence of cluster N6 in HI-injured animals, which was nearly absent in peripheral organs and sham animals (Fig. 6E, F, Supplementary Fig. S9D–F, H). Induction of this phenotype was not caused by the brain-specific isolation procedure, as bone marrow cells treated with the brain immune cell isolation protocol did not change their phenotype (Supplementary Fig. S9I). Compared to peripheral cell clusters, N6 neutrophils were characterized by an activated and mature phenotype (CD11bhigh, CD14high, CD101high, CD44high, Siglec-Fhigh, CXCR2high, CD62Llow) (Fig. 6E). Based on the dichotomous function of early and late brain-infiltrating neutrophils (Figs. 3 to 5), we compared expression of selected markers between day 1 and day 7 after HI. While expression of CD11b and CD45 was similar, a time-dependent phenotypical shift was observed for other markers, the most prominent being an upregulation of CD14 at 24 h and of CD101 at 7 d (Fig. 6G). To identify neutrophil phenotypes, potentially involved in the observed neuroprotective and regenerative effects of late neutrophils, we re-clustered exclusively brain-infiltrating neutrophils from HI mice, resulting in six main subsets (Fig. 6H). Two clusters Br2 and Br3 decreased at day 7 compared to day 1, while cluster Br1 increased 7 days after HI (Fig. 6I). Focusing on the latter, this subset showed an activated aged phenotype (CD101high CD11bhigh CD62Llow CXCR4high CD45high) with a unique expression of Siglec-F. Together, these analyses demonstrated organ- and disease-stage-specific differences between early and late neutrophils with regard to their activation, maturation and function associated with the selective accumulation of Siglec-F positive neutrophils.

Gating on Siglec-F+CD101+ cells (Fig. 7A) and their projection in the UMAP of either all organs (from Fig. 6E) or the brain-specific UMAP (from Fig. 6H) confirmed their selective accumulation in the brain 7 days after HI (Fig. 7B). Comparing this unique Siglec-Fhigh neutrophil population with Siglec-Flow brain neutrophils at 24 h and 7 d after HI, these cells showed a pronounced increase in CD49d and CXCR4 expression while CD62L expression was decreased (Fig. 7C). To investigate the localization of Siglec-Fhigh neutrophils we performed multiplex immunohistochemistry. Interestingly, Siglec-Fhigh neutrophils were found predominantly in the parenchyma, indicating a tissue-imprinted phenotype of this subset (Fig. 7D). To characterize late brain-infiltrating SiglecF+ neutrophils in more detail, we quantified mRNA expression of angiogenesis-related proteins in sorted neutrophil subsets (Supplementary Fig. S9J), revealing a significantly increased expression of *angiopoietin 1, angiopoietin 2* and *VEGFA* in Siglec-F+ cells compared to Siglec-F- brain- and blood-derived neutrophils (Fig. 7E). Based on the selective emergence of Siglec-F+ neutrophils, we then investigated their impact on endothelial migration in scratch assays. Compared to co-culture with Siglec-F- neutrophils, endothelial migration was increased when co-cultured with Siglec-F+ cells, revealed by a 70% increased wound healing compared to untreated cells, which was diminished in the presence of anti-VEGF (Fig. 7F). According to previous studies, showing that particularly GM-CSF induces Siglec-F expression on neutrophils[37], we asked whether the observed upregulation of GM-CSF protein 7 days after HI (Fig. 2A, Fig. 7G) might be related to the emergence of Siglec-F+ neutrophils in the brain. Stimulation of blood-derived Siglec-F- neutrophils with GM-CSF led to a marked increase in expression of Siglec-F, CD11b and CXCR4 (Fig. 7H), similar to the phenotype of late brain-infiltrating Siglec-Fhigh neutrophils (Fig. 7C). Notably, conversion of Siglec-Flow neutrophils to Siglec-Fhigh neutrophils by stimulation with GM-CSF increased neutrophil survival and co-culture with endothelial cells enhanced wound healing responses (Fig. 7I,J).

## Discussion

In the present work, we uncovered unexpected phenotypical and functional neutrophil dynamics in injured brain tissues following neonatal hypoxia-ischemia (HI). Neutrophil infiltration peaked 1 and 7 days after the insult. Both time points overlap with different disease phases, the early one characterized by acute cell damage, whereas at the delayed disease stage regenerative processes are initiated. Reminiscent of these distinct disease stages, early-infiltrating neutrophils aggravated brain damage and late-infiltrating neutrophils contributed to neurological recovery. The disease stage-dependent shift in function from degenerative to pro-regenerative neutrophils was associated with a pronounced alteration of the neutrophil phenotype with early neutrophils being highly activated and late-infiltrating neutrophils displaying a pro-angiogenic phenotype associated with the appearance of a Siglec-Fhigh neutrophil population. In vitro assays implicated that higher cerebral GM-CSF levels at the delayed disease stage might induce this subset, which showed an increased potential for endothelial recovery and enhanced expression of angiogenesis-related genes.

While the majority of studies focused on early, mainly detrimental, effects of neutrophils in ischemic brain injury[2–4], their potential involvement in tissue regeneration at a later stage of disease was elusive. Although neutrophils could be isolated 14 days after ischemic stroke in adult mice, indicating delayed tissue infiltration, their absolute quantity and relevance for pathophysiology remained unclear[20]. In the present study, we demonstrate a biphasic neutrophil recruitment in neonatal HI-induced brain injury, contrasting previous findings in adult stroke with an early infiltration peak 1-3 days and a continuous decline until day 7[7], suggesting different dynamics of inflammatory responses to an ischemic insult in the neonatal and the adult brain[38]. Regional neutrophil distribution showed a disease-stage-dependent accumulation with a high frequency in the thalamus and the meninges at the acute disease stage. Even though these data partially confirm the work by Shrivastava et al., showing a strong infiltration in the hippocampus one week after HI[39], time-dependent regulations in the thalamus differ. About the reasons we can only speculate, but differences in age and experimental models may account for these differences. In addition to meningeal localization of neutrophils, we detected a significant proportion of cells in the brain parenchyma, which, however, strongly depended on the disease stage. While half of brain neutrophils were found in the vasculature at the acute disease stage, supporting an earlier study in a model of neonatal stroke[40], up to 80% of neutrophils were located in the parenchyma or transmigrating from the perivascular space. Despite increased parenchymal localization, late-infiltrating neutrophils were more closely associated with the

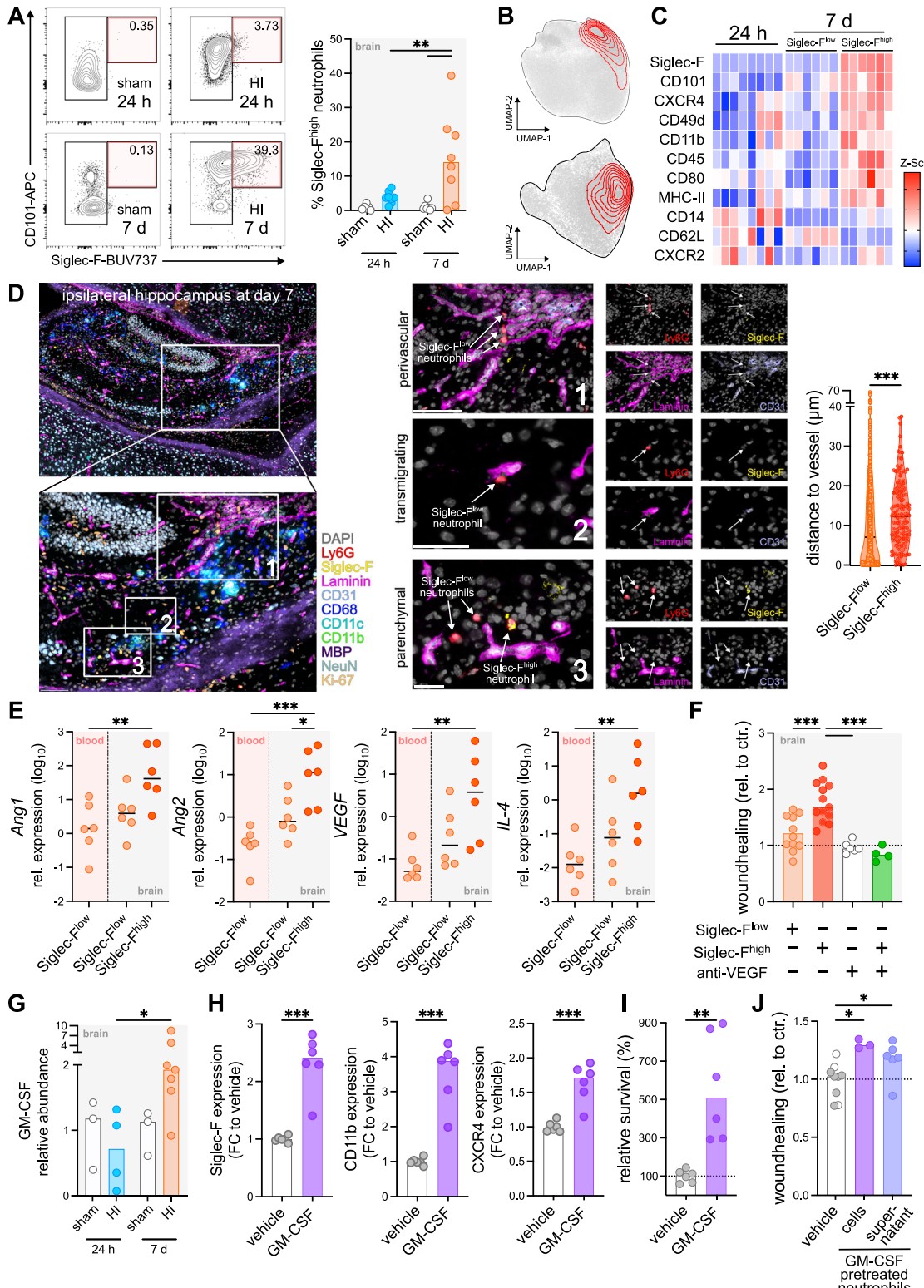

vasculature, which may not solely be related to an active response of late-infiltrating neutrophils considering ongoing developmental vascularization, demonstrated in the present work. In addition to an increased vessel density, the structure of the vasculature differs between both time points with dilated and partially destroyed basement membranes, which might partially explain increased frequency of intraparenchymal neutrophils.

The positive correlation between CD11b expression and neutrophil infiltration suggests a functional interrelation between

intrinsically upregulated CD11b and tissue infiltration. In addition to neutrophil-intrinsic changes, signals from the ischemic brain tissue seem to promote this biphasic infiltration pattern, with osteopontin revealing the most prominent effects. Osteopontin has not only been associated with neutrophil infiltration into infected and injured tissues[41–43], but was also shown to increase CD11b expression on neutrophils[41]. These results suggest that intrinsic differences in neutrophils and organ-specific cues contribute to the observed biphasic recruitment, which is further supported by our ComplexEye assay

**Fig. 7 | Siglec-F$^{high}$ late brain infiltrating neutrophils promote endothelial migration and can be induced by GM-CSF. A** Siglec-F$^{high}$CD101$^{high}$ neutrophils were quantified in the brain, $n = 6$ (sham 24 h), $n = 7$ (sham 7 d), $n = 8$ (HI 24 h/ 7 d). **B** Projection of Siglec-F$^{high}$CD101$^{high}$ neutrophils onto the UMAP of all cells from Fig. 6E or of brain neutrophils from Fig. 6H. **C** Surface marker expression on Siglec-F$^{high}$ compared to Siglec-F$^{low}$ brain-derived neutrophils. Only mice with >2% Siglec-F$^{high}$ neutrophils at day 7 are displayed, $n = 8$ (24 h), $n = 6$ (7 d). **D** Multiplex immunohistochemistry of the ipsilateral hippocampus at day 7. Distance of Siglec-F$^{low}$ and Siglec-F$^{high}$ neutrophils to the next adjacent vessel was quantified for neutrophils from $n = 3$ mice, two-tailed Mann-Whitney test. Scale bar: 50 μm (**E**) qPCR of *angiopoietin 1 (Ang1), angiopoietin 2 (Ang2), VEGFA* and *IL-4* in isolated neutrophils at day 7, $n = 6$ biological replicates, details are provided in the source data file. **F** Wound healing response in scratch assays of bEND.3 in co-culture with brain neutrophils in presence or absence of 5 μg/ml anti-VEGF. Relative wound-healing compared to untreated controls was quantified, $n = 10$ (SiglecF$^{low}$/SiglecF$^{high}$), $n = 6$ (anti-VEGF w/o neutrophils), $n = 4$ (anti-VEGF + SiglecF$^{high}$ neutrophils) biological

replicates, i.e. wells, each with 10,000 neutrophils from 1-4 mice / well. Details are provided in the source data file. **G** GM-CSF protein abundance in brain tissue lysates, determined in proteome profiler antibody arrays (Fig. 2A), $n = 3$ (sham 24 h/ 7 d), $n = 4$ (HI 24 h), $n = 7$ (7 d). **H** Expression of Siglec-F, CD11b and CXCR4 after incubation of isolated Siglec-F$^{low}$ blood neutrophils with 0.5 μg/ml GM-CSF for 18 h, two-tailed unpaired t-test, $n = 6$ mice. **I** Relative survival of isolated Siglec-F$^{low}$ blood neutrophils after incubation with 0.5 μg/ml GM-CSF for 18 h, two-tailed unpaired t-test, $n = 6$ mice. **J** Wound healing response in scratch assays with GM-CSF pretreated neutrophils or their supernatant, $n = 9$ vehicle (3x incubation with neutrophils (white circles), 6x incubation with supernatant (grey circles)), $n = 3$ incubation with GM-CSF-pretreated neutrophils, $n = 6$ incubation with supernatant of pretreated neutrophils. Bar graphs in A/F/G-J and centers in D/E show median values. A/E/F/G/J: One-way ANOVA followed by Šídák's multiple comparisons test. *$p < 0.05$, **$p < 0.01$, ***$p < 0.001$. Individual values and exact $p$-values are given in the source data file.

results. Increased migratory responses to CXCL2 and enhanced protein expression of this chemokine at the delayed disease stage may have contributed to the second infiltration peak of neutrophils. However, neutrophil infiltration might not only rely on protein expression levels, as reflected at the example of GM-CSF, recently described as a neutrophil chemoattractant[44]. Cerebral GM-CSF protein expression was induced 7 days after HI, while migratory responses of neutrophils to GM-CSF were higher at day 1.

In addition to neutrophil-attracting properties, a large proportion of the identified secondarily upregulated proteins mediate neuroprotective and neuroregenerative effects, including neurogenesis, oligodendrocyte maturation and angiogenesis (Supplementary Table S1). These findings support the concept of endogenously induced tissue regeneration 7 days after the insult, coinciding with the second neutrophil infiltration wave. Interestingly, several of the secondarily upregulated or recovered proteins were reproducibly reported to be expressed by neutrophils, including VEGF, angiopoietin 1, IL-1a, ILra, CXCL10[45] and amphiregulin[46]. These findings indicate that neutrophils may not only be modulated by the changed tissue environment, but that they may also actively participate in post-ischemic reparative processes to support neurological recovery.

Disease-stage dependent changes in the tissue environment were associated with a switch of neutrophils' function and phenotype in injured tissues. In support of our previous work[2–4], the present results show tissue-degenerating functions of early brain-infiltrating neutrophils, revealed by an increased neuronal and oligodendrocyte density following depletion. This was not associated with alteration of secondary apoptosis or a modulation of neurogenesis, suggesting that increased neuronal numbers in mice with early neutrophil depletion resulted from an improved survival of neurons that did not die in the acute disease phase. Since late neutrophil depletion also did not affect secondary apoptosis and neurogenesis, neurons and neuronal precursor cells appear not to be the primary target of late-infiltrating neutrophils. In support of this, neutrophil infiltration at the early disease stage correlated with tissue injury, while at the delayed disease stage no clear association could be found. Similarly, glial activation and myeloid cell accumulation as well as their HI-induced phenotypical alterations were not affected by late neutrophil depletion. These data imply a limited impact of neutrophils on myeloid cell/microglia diversification at the delayed disease phase, not excluding a functional interaction between both cell types at other disease time points as suggested in models of adult stroke[4,47].

Considering the growing number of studies suggesting neutrophils to be important for tissue homeostasis and organ development[48–50], we were particularly interested in the impact of late-infiltrating neutrophils on oligodendrocytes and endothelial cells, as postnatal myelination and vascularization are key for proper brain development. Oligodendrocyte and endothelial cell proliferation are endogenously induced, which was strongly impaired in the absence of late-infiltrating neutrophils, suggesting an important role of neutrophils for recovery. Histological alterations at the sub-acute disease stage translated into long-term differences in neurological function at adolescence. Acute damaging effects of neutrophils result in long-lasting deficits in HI-induced hyperactivity. Persisting alterations in emotional behavior, such as hyperactivity and attention dysfunctions have been described in infants suffering from neonatal hypoxic-ischemic encephalopathy[51,52]. Neonatal HI-induced long-term alterations in anxiety-related behavior were more pronounced in the absence of late-infiltrating neutrophils, suggesting that they are important for preventing the development of this functional deficit. Functional differences coincided with long-term histological brain injury, displayed by increased long-term tissue atrophy and stronger myelination deficits in animals with late neutrophil depletion. Reduced oligodendrocyte regeneration detected 10 days after HI might explain long-term myelination deficits and resulting neurological dysfunction. This is supported by clinical studies showing that long-term changes in myelination are associated with neurodevelopmental deficits[53]. Interestingly, differences between sham and isotype-treated animals were less pronounced, indicating that developmental brain maturation may partially compensate detrimental effects of neutrophils in the acute disease phase with regard to myelination. However, the impact of late neutrophil depletion remained strong until adolescence, highlighting the importance of neutrophils in the sub-acute disease phase when regeneration is initiated.

Functional differences of neutrophils were accompanied by pronounced phenotypical changes depending on the stage of disease. The detrimental function of brain-infiltrating neutrophils at the acute disease phase was previously shown to be associated with a high spontaneous ROS production after brain entry[3]. Comparison of ROS production between early and late brain-infiltrating neutrophils showed a significant reduction at the delayed disease stage, suggesting a less activated cell fate at this time point. However, this may not explain tissue protective and regenerative effects of late brain-infiltrating neutrophils. Angiogenesis is a key reparative process following ischemic brain injury[29]. While recent work demonstrated pro-angiogenic properties of neutrophils under physiological conditions[48], their potential contribution to post-ischemic angiogenesis is poorly understood. Interestingly, in vivo and in vitro experimental approaches of the present work revealed pro-angiogenic properties of late brain-infiltrating neutrophils, which was associated with an increased frequency of VEGFR1$^+$CD49$^+$ neutrophils, previously described as pro-angiogenic neutrophils[34,35]. Neutrophils have been suggested to be involved in vascular remodeling in various disease entities, e.g. oxygen-induced retinopathy and sterile liver inflammation[16,30]. In support of this, we show that late-infiltrating neutrophils promote vascular regeneration, especially of small capillaries.

Even though we did not observe detrimental effects by our short-term neutrophil depletion approaches on physiological body and brain development in sham-animals, intervention strategies targeting the total cell population need to be treated with caution. This seems especially important in view of the second neutrophil infiltration peak, uncovered in the present study. Since neutrophil counts recovered 7 days after early depletion, delayed-infiltrating cells appear to be an independent cell population with different function. Nevertheless, characterization of reappearing neutrophils after early depletion showed a slightly more immature phenotype at the delayed disease stage. Keeping in mind the study by Sas et al., reporting an immature neuroprotective neutrophil subset in the setting of adult inflammation-induced optic nerve and spinal cord injury[14], the reappearing immature neutrophil subset might be responsible for the protective effect after early depletion. However, newly appearing neutrophils do not show this neuroprotective phenotype, i.e. elevated CD14 expression. Furthermore, our previous findings showed a strong neuroprotective effect of acute neutrophil depletion already 2 days after HI before reappearance of neutrophils[3]. Although they are slightly different, reappearing neutrophils seem to acquire similar regenerative functions as in isotype-treated animals, e.g. promoting oligodendrocyte and endothelial proliferation and improving long-term myelination. Nevertheless, to exclude potential confounding effects by antibody-mediated depletion, alternative transgenic mouse models, allowing timed depletion of neutrophils[54] will be necessary in future studies to verify the present findings.

Brain-infiltrating and peripheral neutrophils demonstrated pronounced differences in their phenotype, highlighting the enormous plasticity of neutrophils acquiring organ-specific properties[48,55,56]. The most prominent cell cluster in the bone marrow and spleen revealed an immature phenotype, while the most abundant cluster in the blood showed a more mature phenotype. Similarities between spleen and bone marrow neutrophils suggest that the spleen can serve as a niche for hematopoiesis during development[57]. Nevertheless, compared to the bone marrow, the spleen contained another immature neutrophil cluster, not present in other organs, implicating that the spleen as a unique organ regarding neutrophil maturation and subset heterogeneity[58–60]. In the context of ischemic brain injury, disease-stage- and age-dependent changes were described for brain neutrophils[20,61]. In line with the present findings, recent scRNA sequencing analyses in adult ischemic stroke models showed an enrichment of a CD101high neutrophil subpopulation 14 d after the insult, while the dominant neutrophil population at 2 d was characterized by high CD14 expression[20]. Similarly, when comparing blood neutrophils from young and aged animals, a subset of neutrophils with low CD101 and high CD14 expression was associated with worse disease outcome[61]. Therefore, accumulation of CD101highCD14int neutrophils 7 days after neonatal HI cannot solely be explained by developmental changes. Instead, CD101 and CD14 may be used to differentiate between beneficial and detrimental neutrophil subtypes, which needs to be clarified in future studies.

The regenerative properties of late-infiltrating neutrophils coincided with the accumulation of a distinct neutrophil subset with high Siglec-F expression. Similar to the present results, heart-infiltrating neutrophils in myocardial infarction acquired a unique Siglec-F signature, being absent in bone marrow and spleen[62]. Furthermore, Siglec-F+ neutrophils have been characterized as delayed-infiltrating neutrophils in this disease setting, although it remains unclear whether their presence is favorable or not[63]. Interestingly, tumor-promoting SiglecFhigh neutrophils upregulated expression of angiogenesis-related genes[64] and recent work in a model of olfactory neuroepithelial inflammation revealed an organ-specific Ly6G+Siglec-F+ neutrophil subset expressing neurosupportive genes[65]. In support of these previous studies, we detected a significantly increased expression of angiogenesis-related genes in SiglecFhigh brain neutrophils.

Furthermore, compared to Siglec-Flow brain neutrophils at 24 h and 7 d after HI, SiglecFhigh cells showed an increased expression of CD49d and CXCR4, both expressed on angiogenesis-promoting neutrophils[34,35]. Additionally, Siglec-Fhigh neutrophils displayed low CD62L and high CD11b levels, which in combination with high CXCR4 levels resemble the phenotype of aged neutrophils[66,67]. However, their role in brain ischemia is still not fully understood. While a higher proportion of aged blood neutrophils was associated with worse clinical outcome[68], transgenic mice with constitutively aged neutrophils in all compartments had similar infarct sizes compared to controls[66].

Functional characterization of Siglec-Fhigh brain neutrophils demonstrated an increased wound healing response in co-culture with endothelial cells, which was abrogated in the presence of anti-VEGF, suggesting this growth factor being at least partially involved in the regeneration-promoting function of Siglec-Fhigh cells. Interestingly, the phenotype of the delayed emerging neutrophil subset in the brain could be induced in vitro from Siglec-Flow neutrophils by stimulation with GM-CSF. Regenerative effects were also observed in endothelial cell cultures with the supernatant of GM-CSF-treated neutrophils, indicating that direct cell contacts are not needed, which is supported by the fact the Siglec-Fhigh cells were found more frequently in the parenchyma than Siglec-Flow cells. The Siglec-F-inducing effect of GM-CSF has been has been described before in a model of model of renal fibrosis[37] and neuroprotective effects of GM-CSF are increasingly acknowledged in neurodegenerative diseases[69,70]. However, neuroprotection by GM-CSF was mainly related to modulation of T cell responses, while a potential modulation of neutrophils towards a regenerative phenotype has not been reported before.

With regard to clinical translation, interspecies differences have to be taken into account[71,72]. Nevertheless, we observed an overlap of 81% of proteins expressed in adult human and mouse neutrophils, the majority of orthologous proteins involved in immune response pathways[71]. Considering a larger proportion of neutrophils in humans specifically in newborns compared to rodents and adults, it will be important to follow up our experimental findings in the clinical setting of neonatal encephalopathy (NE). This is supported by observational studies in newborns with NE, revealing that disease-stage specific decreases in serum VEGF and GM-CSF levels were associated with abnormal MRI and worse outcomes[73,74]. Though supporting our findings, these data were derived from blood samples and analyses time points can hardly be translated to rodents. Further longitudinal clinical studies with samples from the cerebrospinal fluid combined with comprehensive blood parameter analyses in rodent HI-models might bridge the gap to clinical translation.

Collectively, the present results suggest that neutrophils are heterogeneous between organs and acquire a disease- and time-dependent phenotype in the injured brain. Early brain-infiltrating neutrophils display a hyperactivated phenotype with high CD14 expression and ROS production while late neutrophils acquire an angiogenic and regenerative phenotype with high expression of Siglec-F, which express higher levels of angiogenic factors and which can be induced by GM-CSF. Nevertheless, further studies will be needed to determine the exact molecular mechanisms driving the time-dependent tissue imprinting and verify the functional relevance of the identified neutrophil subset in vivo.

## Methods

### Animal care and group allocation

Experiments were performed in accordance with the Animal Research Reporting of in Vivo Experiments (ARRIVE) guidelines with governmental approval by the State Agency for Nature, Environment and Consumer Protection North Rhine-Westphalia (approval numbers: G1773/20, G1778/20). C57BL/6JOlaHsd mice were initially obtained from Envigo (Netherlands) followed by in house breeding. CatchupIVM mice were recently developed in the laboratory of M.

Gunzer[21] followed by in house breeding. Mice were housed in specific pathogen free individually ventilated cages, kept under a 12-h light/dark cycle with food and water ad libitum. Room temperature and relative humidity were maintained between 20–24 °C and 45–65%, respectively. Body weights of pups were recorded at postnatal day 9 (P9), P10, P11, P12 and P16 and weekly after weaning. A total of 708 C57BL/6 mice ($n$ = 344 female, $n$ = 362 male) and 54 Catchup[IVM] mice ($n$ = 27 female, $n$ = 27 male) derived from a total of 108 litters were enrolled. For all analyses, female and male animals per litter and experiment were randomly assigned to treatment groups. To control for the potential influence of weight and sex, a stratified randomization was performed followed by simple randomization within each block to assign pups to individual groups. All data are presented disaggregated for sex in the source data file. Individuals involved in data analysis knew the animals' designation but were blinded to group assignment. Out of 580 mice ($n$ = 292 female, $n$ = 288 male) exposed to HI, 57 ( = 9.8%, $n$ = 30 female, $n$ = 27 male) died with random distribution between experimental groups. Details about group allocation and mortality rates are provided in Supplementary Table S2.

## Neonatal hypoxia–ischemia

Hypoxic-ischemic (HI) brain injury was induced as previously described[3]. In brief, nine-day-old mice were exposed to HI through cauterization (high-temperature cauter, 1200 °C, Bovie, USA) of the right common carotid artery under isoflurane anesthesia (1.5–4 Vol%), followed by 1 h hypoxia (10% O₂) in an airtight oxygen chamber (OxyCycler, USA) after 1 h recovery with their dams. To maintain nesting temperature, mice were placed on a warming mat during hypoxia[28]. Sham animals received anesthesia and neck incision only. Perioperative analgesia was ensured by subcutaneous administration of 0.1 mg/kg buprenorphine. At the end of the experiments animals were deeply anesthetized with chloral hydrate (200 mg/kg) prior to perfusion.

## Neutrophil depletion

Following previous reports, neutrophils were depleted using an adapted depletion protocol combining anti-Ly6G with anti-rat IgG[26]. Specifically, 5 µg/g body weight anti-Ly6G (clone 1A8, BioXCell, USA) were administered intraperitoneally (i.p.) in 10 µl/g body weight phosphate buffered saline (PBS) either 12 h/36 h/50 h (early depletion) or 6/7/8 days (late depletion) after HI. Anti-rat kappa immunoglobulin (5 µg/g body weight, clone MAR 18.5, BioXCell) was administered 1-2 h after the first and the third anti-Ly6G treatment. Control animals received i.p. injections of isotype control antibody (rat IgG2a, clone 2A3, BioXCell).

## Long-term functional outcome

Behavioral testing was performed on postnatal days 44 and 45, i.e. 5 weeks after HI as previously described[28]. Briefly, animals were transferred to an inverted light/dark cycle after weaning. Behavioral testing was performed during the dark phase in a low noise environment (behavioral unit). The animals' behavior was recorded by using an automatic tracking system (Noldus Ethovision XT (version 15.0, Germany). All test devices were cleaned with 70% ethanol between each trial to minimize interference from odorants. Spontaneous activity and anxiety-related behaviors were assessed in the Elevated Plus Maze (EPM) on the first day followed by the Open Field (OF) test on the second day. In both tests, mice were placed in the center of the maze and behavior was recorded for a duration of 5 min. The time spent in the open arms of the EPM; mean velocities and percent mobility in the OF were quantified.

## FITC hydrogel perfusion and brain tissue clearing

Euthanized animals were transcardially perfused with ice-cold PBS, followed by perfusion with ice-cold 4% paraformaldehyde (PFA) in PBS followed by perfusion with hydrogel containing fluorescein isothiocyanate (FITC)-conjugated albumin according as previously described[31,32]. Subsequently, mice were submerged in ice water for at least 10 min to ensure polymerization of the hydrogel before brains were removed and incubated in 4% PFA in PBS at 4 °C overnight. For visualization of vessel structures in wildtype mice, clearing of brain tissues was performed using a modified iDISCO clearing protocol according to our previous studies[31,32]. Briefly, brain tissues were incubated in tetrahydrofuran (THF, Sigma-Aldrich, USA) with increasing concentrations (30%, 60%, 80%, and 2x 100%, 12 h each) at room temperature with constant agitation at 300 rpm on a horizontal shaker. Samples were cleared in ethyl cinnamate (ECI, Sigma-Aldrich) for at least 12 hours followed by image acquisition in ECI. For simultaneous visualization of neutrophils, Catchup[IVM][21] with tdTomato expressing neutrophils were used. Since signals of fluorescent proteins were reported to fade with the described iDISCO protocol[75], we applied an alternative clearing method with an adapted CUBIC protocol[76] using CUBIC-L (10% (wt/wt) N-butyldiethanolamine, 10% (wt/wt) Triton X-100; TCI, Japan) for delipidation and CUBIC-R+ (45% (wt/wt) antipyrine, 30% (wt/wt) N-methyl- nicotinamide, 0.5% (vol/vol) N-butyldiethanolamine; TCI) for refractive index (RI) matching. First, brains were immersed in ½ CUBIC-L in ddH20 for 24 h and then transferred to pure CUBIC L for 5-6 days until complete decolorization and delipidation with replacement of CUBIC-L solutions after 24 h and then every 2-3 days. Samples were washed three times in PBS for 1 h before transfer to ½ CUBIC-R+ in ddH20 for 3 h. Afterwards, samples were transferred to pure CUBIC R+ for complete RI matching until image acquisition in a 50%/50% (vol/vol) mixture of silicon and mineral oil (Sigma-Aldrich). Confocal and two-photon imaging of brains with intact skulls required adaptations regarding incubation times. Following fixation, the samples were washed in PBS (4 °C, 8 h) and subsequently cleared in the dark on an orbital shaker (75 rpm), employing the above combination of CUBIC-L and CUBIC-R solutions. The skulls were incubated in 50% CUBIC-L (in ddH₂O) at room temperature for 8 h before being transferred to 100% CUBIC-L at 34 °C for 5-8 days. The CUBIC-L solution (20 mL per sample) was refreshed every 24 h. The samples were washed 2-3 times in PBS, each for 2 h at room temperature before transfer to 50% CUBIC-R (in ddH₂O) for 8 h. Finally, the brains with intact skull were immersed in 100% CUBIC-R for refractive index matching at room temperature for 1-2 days prior imaging.

## Light sheet and two-photon imaging

Brains were scanned using a commercial light sheet microscope (Blaze, Miltenyi Biotec, Germany) equipped with 1.1× (0.1NA), 4X (0.35NA) and 12X (0.53NA) objectives with bidirectional light sheet illumination and an sCMOS camera having a 2560 × 2160 chip of 6.5 µm pixel size. Serial optical imaging was performed using the 4X objective acquiring images in ventral-dorsal direction with excitation of FITC at 488 nm and detection of emission signals with a 525/50-nm band-pass filter. For visualization of the vasculature total ipsilateral hemispheres were acquired at ×6.4 magnification with 3 µm steps in the axial direction. For simultaneous visualization and quantification of tdTomato expressing neutrophils total brains were imaged by serial optical imaging with a step size of 5 µm and additional emission detection with 620/60 nm bandpass filter. Analyses of hippocampal vessel densities was performed in 306 ×306 x 900 µm regions of interest (ROIs). Image stacks were preprocessed using ImageJ (version 1.53/1.54, NIH), performing a rolling ball background subtraction (radius, 20 µm) followed by automated analyses using the recently published image analysis pipeline VesselExpress[31,32], an automated open-source pipeline that integrates steps of image segmentation, skeletonization and graph analysis. Vessel length and branching point densities were quantified. A correction factor of 1.318 was applied to compensate for tissue shrinkage due to dehydration and clearing with THF/ECI. For quantification of neutrophil infiltration, the semi-automatic spot function in

IMARIS (version 9/10, Oxford Instruments, UK) was used. To quantify extravasated neutrophils only, spots within vessels were removed using the surface function on the FITC channel followed by manual curation of spots to exclude signals due to non-neutrophil signals. Autofluorescence and vessel signals were used to manually identify and separate hemispheres and to determine hemisphere volumes. Tissue swelling caused by the Cubic clearing protocol was corrected with a length correction factor of 0.858.

Sequential confocal and two-photon imaging of optically cleared brains with intact skulls was performed using the Leica TCS SP8 confocal microscope with Multi Photon (MP) excitation and the Leica Application Suite (LAS X) software (Version 3.1.5.16308; Leica Microsystems GmbH, Wetzlar, Germany). The MP Chameleon Vision II Titan-Saphire laser with an excitation wavelength tuned to 960 nm was used to identify skull and meningeal structures through the second-harmonic generation (SHG) signal (photomultiplier tube bandpass emission filter 460/50 nm). The same region was imaged in a combined second sequence for detection of tdTomato expressing neutrophils and the FITC-labelled vasculature using the confocal argon ion laser (65 mW) excitation at 488 nm for FITC and DPSS (20 mW) excitation at 561 nm for tdTomato. FITC emission was detected with a 500/550 nm PMT BP emission filter and tdTomato emission at 610-700 nm (hybrid detector). Using a 25x objective (HCX IRAPOL x25/0.95 water, Leica Microsystems GmbH) with x0.75 zoom factor and a scan speed of 200-400 Hz resulted in an image size (x/y) of 590,48 x 590,48 μm in a 1024 x 1024 pixels format per single FOV. To capture larger regions, tile scans were performed and subsequently aligned employing the IMARIS stitcher function (Oxford Instruments, UK). The z-stack range reached from 300 to 1200 μm with a z-step size of 2 μm. In images with >500 μm z-range linear z-compensation of the excitation gain was applied. Acquired images were 3D-rendered and processed using IMARIS (version 10.1.0).

### Confocal imaging of cleared brain tissues and quantification of neutrophil-vessel distances

Cleared brains were cut into 3 mm thick slices using a mouse brain matrix (Zivic Instruments, USA). Slices were imaged in a chambered coverslip (ibidi) in CUBIC R+ using an A1plus confocal microscope (Eclipse Ti, with NIS Elements AR software version 4.20, Nikon, Germany). ROIs of 1252.72 μm × 1252.72 μm × 500 μm in the hippocampal region of both hemispheres were imaged with a z-step distance of 1 μm. In IMARIS, bleed-through of the FITC signal into the tdTomato channel was corrected using the channel arithmetic function of IMARIS and neutrophils identified using the spot function. Distances between neutrophils and the next adjacent vessel were calculated using the surface function followed by the distance transformation algorithm in IMARIS.

### MACSima Imaging Cyclic Staining (MICS)

Multiplex immunohistochemistry was performed on 10 μm frozen brain section of animals at day 7 after HI using a MACSima imaging system (Miltenyi Biotec, Germany). This included cyclic immunofluorescence imaging consisting of repetitive cycles of staining with fluorophore-conjugated antibodies, washing, multi-field imaging and photobleaching-based signal erasure. MACSwell sample carriers (Miltenyi Biotec) were mounted on slides containing 4% PFA-fixed brain slices followed by incubation in blocking buffer containing 5% normal horse serum for 1 h at room temperature before sections were pre-incubated with a polyclonal rabbit anti-mouse Laminin at 4 °C overnight. Nuclei were counterstained with DAPI and samples placed in the MACSima imaging system, followed by incubation with a FITC conjugated anti-rabbit antibody in the first cycle to identify laminin. Tissue sections were further stained by step-wise incubation with directly fluorochrome-conjugated antibodies (Supplementary Table S3), each for 1 h at room temp. Acquired pictures were stitched and preprocessed using MACS iQ View Analysis Software (Miltenyi Biotec) and representative overlay pictures are displayed. Cells were segmented based on DAPI positive signals using the StarDist plugin[77] in ImageJ (NIH) and the donut algorithm in MACS iQ View. Siglec-F$^{low}$ and Siglec-F$^{high}$ neutrophils were identified based on their mean expression of Ly6G and Siglec-F and the vasculature was identified by Laminin positive signals. Distance of Siglec-F$^{low}$ and Siglec-F$^{high}$ neutrophils to the vasculature was quantified using MACS iQ View (version 1.3.2).

### Proteome profiler antibody array

To determine the expression of a broad set of proteins at different time points mice were transcardially perfused with PBS 1, 3 and 7 days after sham-operation or HI. Brain sections of 200 μm thickness were dissected from ipsilateral hemispheres between -2.0 mm and -2.3 mm from bregma (hippocampal level) and homogenized in 200 μL radio-immunoprecipitation assay (RIPA) lysis buffer (Sigma Life Science, USA), containing protease and phosphatase inhibitors (complete7, Roche, Switzerland) and 100 mM phenylmethylsulphonyl fluoride (PMSF, Sigma-Aldrich). Samples were incubated on ice for 20 min and centrifuged at 17000×g at 4 °C for an additional 20 min. The supernatant was collected, and the protein concentration determined using the Pierce bicinchoninic acid assay (BCA)-Protein Assay (Thermo Scientific, Germany). 300 μg of isolated protein were analyzed according to the manufacturer's instruction of the Proteome Profiler Mouse XL Cytokine Array (R&D Systems, USA). For visualization and relative densitometric analysis the ChemiDocXRS+ imaging system (Bio-Rad, Germany) and ImageJ software (1.53/1.54, NIH, USA) were used. Mean intensities of protein signals were determined after subtraction of local background intensities. For quantification, values of HI-animals were normalized to the mean value of sham animals from the same time point. Values below background signals were excluded. Differentially expressed proteins (Fig. 2A, Supplementary Fig. S2B) were identified based on mean fold change values above 2 and below 0.5 and by ANOVA analyses (with pooled normalized sham values, including only groups where at least 2 sham and 4 HI values/time point were detectable). HI-induced changes were determined via one sample t-Tests/Wilcoxon tests per time point (Fig. 2A, B). Differences between time points were assessed with ANOVA post hoc tests (Supplementary Fig. S2B). Source data are provided in the source data file.

### Histology

Following behavioral testing, mice were deeply anesthetized with chloral hydrate and transcardially perfused with ice-cold PBS. Brains were removed and snap frozen on dry ice. Tissue injury was assessed on cresyl-violet-stained 20 μm cryostat sections. Tissue atrophy was determined by measurement of intact areas in ipsi- and contralateral hemispheres at a distance of 400 μm using ImageJ software (1.53/1.54, NIH, USA). Volumes were calculated between −1.0 and −2.6 mm from bregma. Tissue loss was determined by comparison with contralateral volumes according to the following equation: 100 − (volume ratio (left vs. right)) × 100.

### Immunohistochemistry

Immunohistochemistry analyses was performed 10 days after HI on 20 μm cryostat sections taken at the hippocampal level (−1.7 to −2.0 mm from bregma). For neuronal and oligodendrocyte densities, neuronal nuclei (NeuN) and oligodendrocyte transcription factor 2 (Olig2) were stained, respectively. Astrogliosis and microglia accumulation were evaluated by staining of glial fibrillary acidic protein (GFAP) and ionized calcium-binding adaptor protein-1 (Iba-1). Endothelial and oligodendrocyte proliferation were assessed in Ki67 immunostainings combined with cluster of differentiation 31 (CD31) or Olig2, respectively. Oligodendrocyte maturation was evaluated by co-staining of Olig2 with adenomatous polyposis coli, clone CC1 positive (referred as CC1, mature oligodendrocytes). For neutrophil

localization in relation to the vasculature, co-staining for CD31, pan-Laminin and anti-Ly6G was performed. Neural precursor cells were stained for doublecortin (DCX). Detailed information on antibodies is provided in Supplementary Table S4. Staining of tissue sections was performed according to previously published protocols[3,78]. Briefly, sections were thawed at 37 °C for 15 min followed by fixation in 4% paraformaldehyde (PFA) for NeuN, Olig2 (host rabbit), CC1 or in ice-cold acetone/methanol for CD31, Ki67, Olig2, DCX, Laminin, anti-Ly6G each for 5 min. Cellular degeneration was assessed by staining of DNA fragmentation using terminal transferase dUTP nick end labeling (TUNEL) according to the manufactures' protocol (In situ Cell Death Detection Kit, Roche, Switzerland). For Iba-1 and GFAP staining, sections were incubated with 4% PFA overnight at 4 °C, followed by antigen retrieval in sodium citrate buffer (10 mM tri-sodium citrate, 0.05% Tween-20; pH 6.0) at 100 °C for 30 min. Nonspecific antibody binding was blocked by incubation with 1% bovine serum albumin, 0.3% cold fish skin gelatin (Sigma Aldrich), 0.2% Tween-20 in PBS for 1 h at room temperature followed by primary antibody incubation overnight at 4 °C. Antibody binding was visualized by incubation with appropriate anti-rat/mouse/rabbit Alexa Flour 488, Alexa Flour 555 and Alexa 647 conjugated secondary antibodies (all: 1:500, Thermo Scientific) for 1 h at room temperature. Nuclei were counterstained with 4',6-diamidino-2-phenylindole (Dapi, 100 ng/ml; Molecular Probes, USA). For analyses of endothelial tip cells, fixed brains were embedded in 4% agarose and cut into 100 µm free-floating coronal sections using a vibratome (Leica 1200 VT; Leica Biosystems, Wetzlar, Germany). Sections were permeabilized by incubation in PBS/0.25% Triton X-100 for 3 h in room temperature, incubated with 4% donkey serum in PBS/0.05% Triton X-100 for 1 h followed by incubation with anti-CD31 for 48 h at 4 °C. After 3x washing in PBS/0.05% Triton X-100, each for 1 h, sections were incubated with Alexa Fluor 594 conjugated secondary antibody (1:250) at 4 °C overnight followed 3x 1 h washing and mounting in ProLong Gold Anti-fade with DAPI (Thermo Scientific).

Confocal imaging with the 20 × objective (A1plus, Eclipse Ti, with NIS Elements AR software, Nikon) was used to generate z-stacks of 14 µm thickness (2 µm focal plane distance). Software-based (NIS Elements AR software version 4.2) quantification was performed in maximal intensity projection images of 3 ROIs in the hippocampus (CA1, CA2, CA3 for NeuN, CD31, Iba-1 and GFAP) and 4 ROIs in the white matter (corpus callosum, cingulum, deep cortical white matter, external capsule for Olig2/CC1, Olig2/Ki67). Endothelial proliferation and neural precursor cells were assessed in the total hippocampal region of stitched large-scale images. Neuronal and vessel densities, astrogliosis and microglia accumulation were quantified by measurement of NeuN, CD31, GFAP and Iba-1 positive areas, respectively. For assessment of proliferating endothelial cells and oligodendrocytes, CD31+Ki67+ and Olig2+Ki67+ cells were counted. Maturation of oligodendrocytes was determined by quantification Olig2+CC1+ cells. TUNEL+ cells were counted in the total hemisphere. Endothelial tip cells were quantified in the hippocampus of z-stack images acquired with a Zeiss LSM 800 confocal microscope through the entire thickness of the tissue section. Endothelial tip cells were defined based on the presence of sprouting filopodia from the end of the cells (Supplementary Fig. S8D) and counted throughout all images of the stacks if they were present in at least two adjacent images.

## RNA and protein isolation

For RNA and protein isolation from tissues, 160-µm-thick tissue sections of the ipsilateral hemisphere were collected at the hippocampal level. RNA and proteins were isolated via the NucleoSpin® RNA/Protein Kit (Macherey–Nagel, Germany) according to the manufacturer's instructions. Isolated RNA was eluted in 40 µl RNAse-free water and quantified using the NanoDrop Spectrophotometer (Peqlab, Germany). Proteins were dissolved in 50 µl Protein Solving Buffer—Tris (2-carboxyethyl) phosphine hydrochloride (PSB-TCEP) provided with the

Kit followed by denaturation at 95 °C. RNA and protein samples were stored at − 80 °C (RNA) and -20 °C (protein) until further processing.

## mRNA expression analysis

mRNA expression in tissue lysates was analyzed according to our previous reports[78]. 1.5 µg of total RNA and TaqMan reverse transcription reagents (Applied Biosystems/Thermo Fisher Scientific) were used to synthesize first strand complementary DNA. Real-time polymerase chain reaction (PCR) was performed in duplicates in 96 well-optical reaction plates for 40 cycles with each cycle at 94 °C for 15 s and 60 °C for 1 min using the StepOnePlus Real Time PCR system (Applied Biosystems/Thermo Fisher Scientific). PCR products of *NeuN, Map2, CC1, MBP, CNPase and MAG* were quantified using assay on demand primers and fluorogenic reporter oligonucleotide probes obtained from Applied Biosystems/Thermo Fisher Scientific (Supplementary Table S5). Validation of TaqMan Assays is published on the website of the company. Ct values were normalized to the housekeeping gene beta-2-microglobulin [Δct = ct (target gene) − ct (beta-2-microglobulin)] and related to the mean of sham animals using the ΔΔCT formula. Fold change values were calculated.

For mRNA expression analyses in brain and blood neutrophil populations, FACS-sorted cells (Supplementary Fig. S9) were stored in 50 µL of RNAlater solution (Invitrogen, Thermo Fisher Scientific, Waltham, MA, US) at -20 °C. RNA was isolated using Qia Shredder and RNeasy Mini Kit (Qiagen, Hilden, Germany) and cDNA transcribed using the Superscript II Reverse Transcriptase Kit (Invitrogen, Thermo Fisher Scientific). Real-time PCR was performed at 60 °C annealing temperature to determine expression for *VEGFA, Angiopoietin 1, Angiopoietin 2* and *IL4. Rps9* served as housekeeping gene. Primer sequences are provided in Supplementary Table S5, which are also published on the website of OriGene (Newark, USA). mRNA expression was measured using the Luna Universal qPCR (New England BioLabs, Ipswich, MA, US) with SYBR® Green. Relative gene expression was calculated with 2^-ΔCt formulation.

## Western blot analysis

Protein concentration was quantified by using the Protein Quantification Assay (Macherey–Nagel, Germany), followed by protein separation on 12.5% SDS polyacrylamide gels including 2,2,2-trichlorethanol for visualization of total protein abundance with ultraviolet light. Fluorescent bands of the total protein on the gel were captured using the ChemiDoc Imaging System (Bio-Rad). Separated proteins were transferred to nitrocellulose membranes (0.2 µm, Amersham, USA) at 4 °C overnight. Equal loading of 7.5 µg / lane and transfer of proteins was confirmed by staining of membranes with Ponceau S solution (Sigma Aldrich). Nonspecific binding was blocked by incubation in 5% nonfat milk powder (Cell Signaling, USA), 0.1% Tween-20 in Tris-buffered saline (TBS) followed by incubation with mouse anti-MBP (1:3000, Convance, USA) each in blocking solution at 4 °C overnight. Membranes were incubated with anti-mouse peroxidase-conjugated secondary antibody (1:5000, DAKO, Denmark) in blocking solution at room temperature for 1 h followed by chemiluminescent detection with the enhanced chemiluminescence prime western blotting detection reagent (Amersham, GE Healthcare Life Science, USA). For visualization and densitometric analysis, the ChemiDocXRS + imaging system and ImageLab software (version 5.1, Bio-Rad, Germany) were used. MBP protein expression was normalised to the total protein load (Supplementary Fig. S6). The total of 54 samples was equally divided to different gels/blots, each containing isotype-treated sham samples serving as controls. For quantification, ratios between the target protein (Supplementary Fig. S6C) and the total protein amount (Supplementary Fig. S6A) were normalized to the mean of sham control samples loaded on the same gel. Original images of full length blots and gels used for representative images in Fig. 3H are provided in Supplementary Fig. S6.

## Isolation of single cell suspensions and cell sorting, conventional flow and spectral flow cytometry

Mice were euthanized by i.p. injections of an overdose of chloralhydrate followed by transcardial perfusion with ice-cold PBS and removal of spleens, femur/tibia and brains. Blood specimens were collected with ethylenediaminetetraacetate (EDTA) coated capillaries (CLINITUBES, Radiometer, Germany) by snipping the right atrium of the heart immediately prior to perfusion via the left ventricle and transferred into EDTA coated collection tubes (Minicollect, Greiner Bio One, Germany). Ipsilateral brain hemispheres and spleens were dissected and homogenized through a 70 μm cell strainer (BD Biosciences) by continuous rinsing with 15 ml of ice-cold hanks buffered saline solution (HBSS, Gibco, Thermo Scientific) supplemented with 0.06% bovine serum albumin (Sigma-Aldrich), 0.6 mM EDTA (Carl Roth, Germany), pH 7.4. Erythrocytes in spleen and blood samples were lysed by incubation with RBC lysis buffer (BioLegend, USA) for 1 min (spleen) and 5 min (blood) on ice followed by a washing step with HBSS. Neonatal femurs and tibiae were pre-shredded before the bone marrow was homogenized through a 70 μm cell strainer by continuous rinsing with 15 ml of ice-cold HBSS. Homogenized brain samples were centrifuged at 400×g for 10 min at 18 °C and the supernatant was discarded. The pellets were resuspended in 7 ml 37% Percoll (Sigma Aldrich, Germany) in 0.01 N HCl/PBS and centrifuged at 2800×g for 20 min at 18 °C. The cell pellet was washed in HBSS prior to staining in Cell Staining Buffer (BioLegend, for spectral flow cytometry) or HBSS (for conventional flow cytometry and sorting) for 20 min at 4 °C (Supplementary Table S6). Viable Siglec-F$^+$ and Siglec-F$^-$ CD45$^+$CD11b$^+$Ly6G$^+$ neutrophils (Supplementary Fig. S9J) were sorted with the BD FACSAria™ Cell sorter (BD Biosciences).

## Conventional flow cytometry and quantification of ROS

Isolated cells were incubated with antibody cocktails (Supplementary Table S6) in for 20 min at 4 °C. For quantification of brain-infiltrating neutrophils, doublets were excluded and viable neutrophils were identified as CD45$^{high}$, FvD$^-$ (fixable viability dye, ebiosciences, Germany), CD11b$^+$ and Ly6G$^+$ cells (Fig. 1A, Supplementary Fig. S1A). Total cell counts were determined using BD TrueCount beads (BD Biosciences). The proportion of pro-angiogenic CD49d$^+$VEGFR-1$^+$ neutrophils was quantified according to gates set with fluorescence minus one (FMO) controls (Supplementary Fig. S9A). Another cohort of mice was used to determine ROS production in brain-infiltrating neutrophils by staining with 2.5 μg/ml Dihydrorhodamine 123 (Sigma-Aldrich) for 15 min at 37 °C followed by 15 min incubation on ice and one washing step prior to measurement. Mean fluorescence intensities (MFI) were quantified according to FMO controls. Neutrophil depletion efficiency was quantified by intracellular staining with anti-Ly6G using the eBioscience™ Foxp3/Transcription Factor Staining Buffer Set (Invitrogen, Germany) according to the manufacturers' instruction. Detailed information on antibodies is provided in Supplementary Table S6. Data acquisition and analyses were performed with BD FACS LSRII equipped with FACS Diva software (version 6.1.3, BD Biosciences).

## Spectral flow cytometry of neutrophils and myeloid cell types

Isolated cells were incubated with the indicated mixtures of fluorophore-conjugated antibodies (Supplementary Tables S7 and S8) for 20 min on ice followed by a washing step prior to measurement. Doublets were excluded using forward and side scatter characteristics and dead cells were excluded using DAPI. Data acquisition was performed on a 5L-Cytek® Aurora (Cytek Biosciences, USA) and data were analyzed using FlowJo (v10.10.0, BD Bioscience) and GraphPad Prism 10 (GraphPad Software, USA). For further analyses, all samples were down-sampled and concatenated to have comparable numbers of neutrophils and macrophages/microglia from all experimental groups. Dimensional-reduction and unsupervised clustering was performed using the

Uniform Manifold Approximation and Projection (UMAP) and FlowSOM plugins in FlowJo. Marker expression of different clusters are displayed as Z-Scores. Data are visualized using GraphPad Prism 10 and R. Neutrophils were characterized as CD45$^+$CD11b$^+$Ly6C$^{int}$CD115$^-$Ly6G$^+$ cells (Supplementary Fig. S9B) and low abundant clusters were merged with similar clusters. For projection of the Siglec-F$^{high}$ CD101$^{high}$ subset onto UMAPs of neutrophils from all organs or from 7d-HI-brain in the concatenate were gated and their distribution in the different UMAP embeddings is displayed (Fig. 7B). Macrophages/microglia were identified as CD11b$^+$TMEM119$^+$P2RY12$^+$ cells (Fig. 4D, Supplementary Fig. S7E).

## ComplexEye Assay

Migration patterns of neutrophils to a variety of chemoattractants and stimulating proteins were analyzed in our previously developed multilens array microscope designed for high-throughput and high-resolution imaging of cell migration[25]. Sorted blood neutrophils from day 1, day 3 and day 7 were collected in Dulbecco's Modified Eagle Medium (DMEM, 4.5 g/L Glucose, w: stable Glutamine, w: sodium pyruvate, w: 3.7 g/L NaHCO3, Pan Biotech, Germany) supplemented with 10% fetal calf serum (FCS, Sigma Aldrich), 1 mg/ml Penicillin / Streptomycin (P/S. Pan Biotech). After centrifugation at 350 g for 5 min cells were resuspended in pre-warmed medium and seeded in 64 wells of a 384 well plate, with 2000 cells / 50 μl / well followed by incubation either with vehicle or with the following stimuli: CXCL1/2/5/12/10/16, CCL6/11, GM-CSF, LTB4, PAF or osteopontin. Detailed information on concentrations and suppliers is provided in Supplementary Table S9. Neutrophil behavior was imaged at a rate of 2 frames per minute (30 s interval) for 2 hours. Resulting movies were segmented, tracked and analyzed via the Napari platform in combination with the MMV_H4Tracks plugin according to our previous work[25]. The percent of moving cells and the mean moving speed were quantified and related to the PBS control value of the same sample.

## Aortic ring assay

The day before cell sorting, aortic rings were prepared as previously described[79]. In brief, thoracic aortae from 12-day old C57BL/6 mice were dissected, fat and connective tissue removed and 0.5 mm aorta rings were placed in 96-well plates coated with type I collagen (Enzo Life Sciences, Farmingdale, USA). Once embedded, the rings were covered with endothelial cell growth medium (Promocell GmbH, Germany). The next day, $2 \times 10^4$ sorted brain neutrophils derived from 2-5 mice were added per ring. Details are provided in the source data file. Nine days after incubation at 37 °C rings were imaged using an AMG EVOS inverted digital phase contrast microscope (Fisher Scientific, Germany) equipped with a Sony ICX285 monochrome CCD-Chip camera. Vascular sprout lengths were measured using ImageJ software (version 1.53).

## Cell scratching and wound-healing assay

Mouse brain endothelial cells (bEnd.3) were kindly provided by Dr. Egor Dzyubenko. Cells were maintained in DMEM, 4.5 g/L Glucose, w: stable Glutamine, w: sodium pyruvate, w: 3.7 g/L NaHCO3 (Pan Biotech, Germany) supplemented with 10% fetal calf serum (FCS, Sigma Aldrich), 1 mg/ml P/S (Pan Biotech). Cells were seeded with 30,000 cells / well in 200 μl medium a 96-well plate. After 48 hours wounds were made with a sterile pipette tip; dislodged cells were washed away with PBS and wounds were imaged with an AMG EVOS inverted digital phase contrast microscope (Fisher Scientific) equipped with a Sony ICX285 monochrome CCD-Chip camera. Afterwards cells were incubated either with 10,000 sorted neutrophils in 200 μl DMEM 10% FCS, 1% P/S or with supernatant or with pretreated neutrophils. Pretreatment of neutrophils with 500 ng/ml GM-CSF (Biolegend) was performed with 50,000 neutrophils in 100 μl in round bottom 96-well plates for 18 h. Wounds were imaged after 18 h in co-culture and the percentage of healed area was calculated. To correct for confounding

effects by differences in endogenous healing speed between different experiments, values with treatment were normalized to non-treated control cells of the same plate and experiment.

## Statistical analyses

Results (unless indicated otherwise) are presented as scatter plots with bars showing median values. Sample size for immunohistochemistry analyses and assessment of long-term functional deficits were determined a priori using G*Power (version 3.1), assuming an effect size f [by ANOVA] of 0.6. An α-level of 0.05 and a power of 0.8 were required. A mortality of 10% was assumed for the present injury model, yielding a final sample size of 10 animals per group. For statistical analysis, the GraphPad Prism 9/10 software package was used. Data were tested for Gaussian distribution with the D'Agostino & Pearson test or in case of small n-numbers with Shapiro-Wilk and Kolmogorov-Smirnov tests. For comparisons of more than two groups, either ordinal two- or one-way ANOVA (parametric) or Kruskal–Wallis (non-parametric) with post hoc Šídák's or Holm Šídák's multiple comparison tests for parametric data and Dunn's multiple comparison tests for non-parametric data were applied. In case two-way ANOVA was required in non-parametric data, values were log-transformed as indicated in the graphs and the source data file. Data for weight development were analyzed by 2-way-repeated measure ANOVA followed by Tukey's multiple comparison test. To compare two groups, either unpaired two-tailed students t-test (parametric) or Mann Whitney-Test (non-parametric) was applied. To determine stimulation responses in ComplexEye assays one sample t-test (parametric) or Wilcoxon-Test (non-parametric) was used. All applied tests are mentioned in the figure legends and detailed results of statistical analyses are provided in the source data file.

## Reporting summary

Further information on research design is available in the Nature Portfolio Reporting Summary linked to this article.

## Data availability

A reporting summary for this article is available as Supplementary Information file. The data supporting the findings of this study are available within the article and its Supplementary Figs.. The source data underlying the main and Supplementary Figs. and details on statistical analyses, including exact *p* values are provided in the source data file. Source data are provided with this paper.

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

## Acknowledgements

We thank M. Rizazad, K. Kempe and I. Spyra for excellent technical assistance. Light sheet microscopy was performed at the Imaging Center Essen (IMCES), a service core facility of the Faculty of Medicine of the University Duisburg-Essen, Germany. We thank A. Brenzel and Dr. A. Squire from the IMCES for technical support in image acquisition and analyses. We thank Dr. Egor Dzyubenko for providing bEnd.3 cells. This work was supported by the Deutsche Forschungsgemeinschaft (DFG) to J.H. (HE-7363/2-1), J.J. (JA-2461/7-1, TRR332 (449437943), project A5), MG (TRR332 (449437943), project C6 GU769/15-1, GU769/23-1), D.M.H. (TRR332 (449437943) project C6), O.S. (TRR332 (449437943) projects A2 & Z1, CRC1009 (194468054) project A13, CRC1123 (238187445) project A6, CRU342 (414847370) project A1) and V.S. (SI-2650/1-1). The work was further supported by funding to O.S. form the Leducq Foundation, the IZKF and the IMF of the Medical Faculty in Münster, and from Novo Nordisk. The work conducted in the Leibniz-Institut für Analytische Wissenschaften – ISAS – e.V. was supported by the "Ministerium für Kultur und Wissenschaft des Landes Nordrhein-Westfalen" and the "Senatsverwaltung für Wissenschaft, Gesundheit und Pflege von Berlin", and was further supported by the Bundesministerium für Forschung, Technologie und Raumfahrt, BMFTR. We thank C. Richter for graphical design of organ icons and experimental paradigms in Figs. 3, 4, 6 and S9.

## Author contributions

J.H. and M.R. conceptualized and designed the study. M.R., E.D., E.P., N.L., C.K., D.N., L.K., A.J.F., M.S., M.J., N.H., E.A.A. and J.H. performed experiments, helped in acquisition and data analyses. R.C., C.J.E., V.S., D.M.H., M.G., J.J., U.F.M., I.B. and O.S. provided significant input in interpretation and discussion of the data. M.R. and J.H. drafted the manuscript and figures. All authors revised and approved the manuscript.

## Funding

## Competing interests

The authors declare no competing interests.

## Additional information

¹Department of Pediatrics I, Neonatology & Experimental Perinatal Neurosciences, University Hospital Essen, University Duisburg-Essen, Essen, Germany. ²Center for Translational Neuro- and Behavioral Sciences (C-TNBS), University Hospital Essen, University Duisburg-Essen, Essen, Germany. ³Institute of Experimental Pathology (ExPat), Center of Molecular Biology of Inflammation (ZMBE), University of Münster, Münster, Germany. ⁴Department of Otorhinolaryngology, Translational Oncology, University Hospital Essen, University Duisburg-Essen, Essen, Germany. ⁵German Cancer Consortium (DKTK) partner site Düsseldorf/Essen, Essen, Germany. ⁶Institute for Experimental Immunology and Imaging, University Hospital Essen, University Duisburg-Essen, Essen, Germany. ⁷Department of Neurology, University Hospital Essen, University Duisburg-Essen, Essen, Germany. ⁸Department of Molecular and Clinical Medicine, Institute of Medicine, Wallenberg Centre for Molecular and Translational Medicine, University of Gothenburg, Gothenburg, Sweden. ⁹Institute of Neuroscience and Physiology, Sahlgrenska Academy, University of Gothenburg, Gothenburg, Sweden. ¹⁰Leibniz-Institut Für Analytische Wissenschaften - ISAS - e.V., Dortmund, Germany. ¹¹These authors contributed equally: Mathis Richter, Eva Diesterbeck. ✉e-mail: josephine.herz@uk-essen.de

