## [Transparent Peer Review file · Nature Communications]

Hypoxic-ischemic brain injury in neonatal mice sequentially recruits neutrophils with dichotomous phenotype and function

Corresponding Author: Dr Josephine Herz

Version 0:

Reviewer comments:

Reviewer #1

(Remarks to the Author)

In the current manuscript Richter and cols. describe neutrophil infiltrating dynamics to the ischemic brain in a HI Neonatal encephalopathy model. They identify an early (24h) and late (7 D) neutrophils waves with different phenotypical and functional roles, associated with activation or angiogenic activity respectively. Depletion of early infiltrating neutrophils is neuroprotective and has long term positive effects on mouse behaviour, whereas later time point depletion impaired vascular and oligodendrocyte regeneration and correlates with long term anxiety-related behavior. Authors made a very intriguing observation identifying how neutrophil heterogeneity mediates different roles during brain recovery and their potential implication in long term neurocognition. There are some key points that should be reinforced:

Major points:

-Neutrophil depletion model: The authors take advantage of a combination of anti-Ly6G and an Anti-Rat IgG. Author indicate that “we obtained a significant reduction of both, circulating and brain-infiltrated neutrophils, which recovered at day 7 (Suppl. Fig. S2A)” However, they do not show which is the depletion efficiency if depletion takes place at day 7. The authors should check whether the depletion protocol combining anti-Ly6G with anti-rat IgG is efficient also with late depletion.

-According to the phenotypic profiling, neutrophils from day 7 seems to be more “mature” (e.g. higher expression of CD101) I wonder if this “maturation differences” impacts on the depletion efficiency of the model. In line with this, previous observations on a model of nerve injury shows a neutrophil subpopulation that express relatively low amounts of Ly6G and Ly6B, and relatively high expression of CD14 that were associated with nerve regeneration (Sas et al., Nat Immunol. 2020), authors enriched this population by using anti-CXCR2 treatment. I believe that an alternative model to deplete neutrophils should be included (E.g. iDTR). This will reinforce the central hypothesis of the article and control better for confounding factors such as the neutrophil maturity status or the long-term effect of the antibodies reaching the brain parenchyma (E.g. behavioural tests).

-Moreover, little is known on the effect of the neutrophil depletion model in other leukocytes (e.g. microglial cells).

Neutrophil subsets behaviour:

-In Figure 3F and 3E, the authors state that neutrophils distribution and proximity to vessels differs at 24h and 7d post HI. In figure 3E, they show a clear difference in endothelial cells (CD31 staining). Could the difference in neutrophils localization to vessels be due to a change in endothelial compartment rather than in neutrophils themselves? How the authors envision a possible implication of localization in their model?

-The authors identify Siglec-F as a marker for late infiltrating neutrophils but lack any functional further characterization. For example, it would be nice to see whether the Siglec-F-high neutrophils have higher angiogenic capacity, their distribution in the damaged brain.

Minor:

1. Figure 1B: the image of 24h shows a substantial number of neutrophils in the contralateral hemisphere. This is in apparent contradiction with quantification by imaging and FACS in Supplementary Figure 1A-B. Please change the image if it is not representative. As well, the scale bar is missing.

2. Figure 2B: the counterbalance of weight loss in early depleted mice is not clear. It might not be significant but clearly, the

trend is the same as in HI mice.

3. Figure 3F: it is not specified at which time point the picture is taken. Show the missing time point to clearly show the difference in localization. Investigate the location and closeness to vessels of Siglec-F positive neutrophils.

4. Supplementary Figure 3C: change the scale of the Y axis in the last two graphs to better see the data.

Reviewer #2

(Remarks to the Author)

I co-reviewed this manuscript with one of the reviewers who provided the listed reports. This is part of the Nature Communications initiative to facilitate training in peer review and to provide appropriate recognition for Early Career Researchers who co-review manuscripts

Reviewer #3

(Remarks to the Author)

The study examines the role of neutrophils in a neonatal hypoxia-ischemia (HI) model in postnatal day 9 mice, a model of hypoxia-ischemia encephalopathy (HIE) in at-term newborns, by testing effects of neutrophil depletion either early (12/36/50h) or late (6/7/8d) after HI on heterogeneity of peripheral neutrophils and neutrophils in injured brain as well as effects of depletion on early injury in the hippocampus and long-term behavior recovery. The authors demonstrate biphasic recruitment of neutrophils during acute injury (1d) and sub-chronic injury (7 and 9d) after HI and show phenotypical and functional diversity of neutrophils during these injury phases. Importantly, they show that neutrophil depletion early after HI protects neurons in the hippocampus whereas depletion at the later time point promotes death of neurons in the hippocampus, adversely affects oligodendrocyte regeneration and vascular bed, and exacerbates long-term behavior outcome, as manifested by enhanced anxiety-related behavior. The purely injurious role of neutrophils in the pathophysiology of cerebral ischemia in adults and in neonates is being reconsidered. Thus, the significance of this study is in detailing the dynamics of spatio-temporal appearance of neutrophils in injured neonatal brain, demonstration of neutrophil heterogeneity after HI injury and evolved neutrophil phenotypes over time, and in relation to injury. Translational potential of the study is in demonstration that targeting neutrophils during later injury phases, not early when a diagnosis is often yet to be made in newborns, can be beneficial long term. At the same time, there are multiple concerns regarding quality of the data and data presentation, which ultimately affect data interpretation. Conceptually, the gold standard while studying/reporting effects of interventions in cerebral ischemia and hypoxia studies is histological outcome, but there are no data on histological outcomes at any point. Furthermore, presentation of several types of results is questionable and, while the authors acknowledge the dynamic nature of changes in the developing brain, including maturation of the blood-brain barrier interphase and leukocytes, they inappropriately lump the data from contralateral hemisphere (i.e., hemisphere directly affected by hypoxia and remotely by ischemia) obtained at from hours to 10 days after HI. There are also a number of misconceptions regarding cell identities. All together, conclusions are not fully supported by the data and rigor criteria are not met.

In detail:

I. The magnitude of neutrophil accumulation over time.

The 3D data on neutrophil presence in injured CATCHUP/IVM pups and some of neutrophil molecules that can affect injury evolution/recovery are novel, meaningful, and enlighten the HI field.

Concerns:

1. The data for contralateral hemisphere are inappropriately lumped for 5 time points, 12h, 24h, 3d, 7d and 10d. Contralateral (hypoxic) hemisphere is known to be affected in the HI model. In fact, the data in Fig 1b show sizable neutrophil accumulation in contralateral hemisphere at 24h.

2. Given biological variability in the HI model, n=4 per time point is insufficient. Typically at least n=8-15 pups per group is utilized. Fig. 1B shows vastly different magnitude of neutrophil accumulation in a total of four mice at 7d, from ~2.5x10⁴ cells in one mouse to ~25x10⁴ cells in another mouse and ~9x10⁴ cells in other two mice. Variability is likely related to varying extent of injury in individual mice. Histology data should be presented for 24h and 7d to show variability in the extent of brain swelling at 24h and the magnitude of tissue loss in the cortex and the hippocampus by 7d.

3. In Fig. 3D neutrophil numbers in contralateral hemisphere should be added.

4. Fig. 1G. What are the data compared to? If they are compared to the data from shams lumped from all time points, more appropriate analysis should be done after separating the data for different time points.

5. In studying effects on injury, including HI, a vastly different animal number per group, like n=4 in experimental group Vs more than 12-15 in control group, is to be avoided.

6. The authors categorically state that neutrophils infiltrate acutely injured HI regions and link this notion to the knowledge in adult stroke models, but it has been documented that neutrophils can signal without entering the parenchyma in stroke and release ROS and other molecules while being adherent to the vasculature. Some studies (Engelhardt) demonstrated that in stroke models the majority of neutrophils accumulate within the neurovascular unit and the subarachnoid space where they remain separated from the brain parenchyma by the glia limitans.

7. Based on appearance of data in Suppl Fig.1, there are 2 subgroups with distinct levels of CCL11, CCL6 and angiopoietins. Was injury pattern/severity same in those mice?

II. Effects of 'early' Vs. 'late' neutrophil depletion on long-term functional outcomes. Long-term effects of neutrophils on brain recovery after HI have never been studied. Thus, the data on how neutrophil depletion during individual injury stages affects long-term outcomes are novel and impactful.

Concerns:

1. Data presentation is confusing. For example, Fig. 2C left shows speed in cm/sec in sham and HI mice (average ~6 cm/sec in shams), whereas on the right Y axes is labeled in cm/sec but the data are presented as normalized to 1, which is inconsistent. More importantly, considering highly variable speed in shams, it is more appropriate to show all data as speed and compare values in neutrophil depleted mice to that in shams and/or HI with intact neutrophils.
2. Histology at 5 weeks must be presented and in reference to speed/mobility.
3. One of studied aspects is the effects on oligodendrocyte accumulation, which is important, but the ultimate outcome of oligodendrocyte survival, proliferation and differentiation is development of the white matter. Incorporating effects on the white matter presence/integrity may delineate an important underlying mechanism of recovery and will certainly strengthen the manuscript.

III. Identification of leukocyte subtypes and their roles.

Concerns:

1. Iba-1 can not discriminate between microglia, freshly accumulated monocytes or differentiated macrophages (Suppl Fig. 3B). Cell type specific markers are to be used to interpret the phenotypes of Iba-1 cells. There is marked accumulation of monocytes that differentiate by 7d in this model.
2. CD11b/Ly6C flow cytometry plots (Suppl Fig 4A) are used to gate on neutrophils (Ly6G-high/Ly6C-intermediate Vs. monocytes (Ly6C-high). There are two subpopulations of cells within the chosen box, but the entire Ly6C+ population is chosen, not Ly6C-intermediate. Given such selection and knowing that CD14 is abundantly expressed by monocytes and macrophages is it possible that many of CD14+ cells are in fact monocytes?
3. There is substantial interplay between neutrophils and monocytes and/or microglia, interactions that largely mediate injury progression. These interactions need to be considered to support conclusions.
4. The phenotypes of neutrophils upon depletion and reappearance can be very different, which would affect interaction with cells of the monocyte lineage. These mechanisms have not been considered.

IV. Neutrophil –mediated protection against sub-chronic injury and effects on angiogenesis.

Overall the data clearly demonstrate protection of hippocampal neurons by the presence of neutrophils (Fig. 3B) and increased oligodendrocyte maturation (CC1+) 10d after HI.

Concerns:

1. Data presentation in this figure suffers the same problem as in Fig. 2, i.e., the number of neurons is highly variable in individual mice with intact neutrophils, thus, expressing the data in mice with depleted neutrophils relative to “1” is uninformative and can be misleading.
2. Based on appearance of Olig2+/CCL1+ and Olig2+/Ki67+ data in Fig. 3C&D there are 2 subgroups. Was injury volume/distribution was similar in individual mice in those subgroups?
3. The results related to vessel coverage are unclear. What was measured in Fig.3E, pixel number? Vessel size distribution, including establishment of capillary networks that occurring during a studied time frame should be reported. Was there measurable number of tip cells?

V. The data in aortic ring assay are elegant but there are not really relevant to the studied effects in the brain as neural vasculature is so distinct in many regards including presence of the BBB.

VI. Sex differences are not considered.

Reviewer #4

(Remarks to the Author)

The manuscript by Ritcher et al and entitled: Neonatal Hypoxic-ischemic brain injury sequentially recruits neutrophils with dichotomous phenotype and function, is a fine study characterising the biphasic influx of neutrophils with distinct characteristics in neonate brains post HI. This is a very informative study that provide some novelties: specifically, the authors' data suggest that early neutrophil influx is detrimental whilst late neutrophil influx is beneficial (associated with angiogenesis), responses associated with their phenotypical characteristics and the selective depletion.

There are however outstanding questions that the authors should consider:

- The authors should provide more details on the mechanisms of recruitment of neutrophils at the different timepoints. Whilst this study gives some insights into the indirect mediators present in the brain during neutrophil recruitment, they have not investigated the classical neutrophils chemoattractants at the two time points of the response (i.e. CXCK1/2, Lix, CXCL12 etc, LTP4, PAF, FPR agonists etc..). Of note, Zhang et al (Front. Neuro 2021) demonstrated the role of GM-CSF in the early recruitment of neutrophils, the current authors have not discussed or investigated this possibility either; and in the light of their different phenotypic characteristics, the authors should provide stronger mechanistic insight into the recruitments of the neutrophils rather than classical inflammatory mediators.

- Whilst the authors showed a clear influx of neutrophils in the brain at day 24 and D7, it would be informative if they could give more details on the location of those neutrophils using higher resolution images: are they solely peri or intravascular? Do they locate in a specific structure of the brain, same for both influx or in different locations? The images provided are unclear. Of note, the study by Shrivastava et al (neuro Res Int 2012, not cited) demonstrated already the predominance of neutrophils in the neonatal brain in specific regions at day-7 that is different from the neutrophil influx at the early time-point post HI.

- What are the functional characteristics of late-infiltrated neutrophils? the authors provided details on their phenotypical characteristics and the effect of their depletion, but one could argue that the tissue vascularisation response for the later in

an indirect phenomenon. For instance, are the pro-angiogenic neutrophils responsible for the high VEGF detected at 7 days (if so, they could consider using VEGF conditional KO)? Or is it through a different molecular mechanism (e.g. IL-4??) Depletion is insufficient to demonstrate the functionality of those cells (as this could be an indirect response due to the lack of their presence or death in the tissue etc..).

- Regarding the effect of the neutrophil depletion on neurons, is the changes in neuronal cells due to lack of proliferation or apoptosis?

We are confident that if the authors answer those simple questions, the manuscript will be greatly strengthened.

Version 1:

Reviewer comments:

Reviewer #1

(Remarks to the Author)

Authors have addressed the main concerns raised in my revision and additional experiments included reinforce the conclusions of this study.

Reviewer #2

(Remarks to the Author)

Reviewer #4

(Remarks to the Author)

I thank the authors for providing clear answers to my comments and adding the new results to their original manuscript.

Reviewer #5

(Remarks to the Author)

Thankyou for this excellent manuscript and very comprehensive well-written overview of this important area.

Minor questions:

Can you mention interspecies differences in neutrophil counts and function if relevant to explain the gap in knowledge to date and the further relevance of your research? In view of the fact that in mice only 20% of leukocytes are neutrophils and this is 80% in humans this may imply that your research may be even more relevant in human infants.

It would be helpful to link with some of the human data on VEGF, GFAP and GMCSF in the relevant patient group

We thank the reviewers for their very careful evaluation and constructive criticism to improve scientific rigor and overall quality of our manuscript.

Reviewer #1 (Remarks to the Author):

In the current manuscript Richter and cols. describe neutrophil infiltrating dynamics to the ischemic brain in a HI Neonatal encephalopathy model. They identify an early (24h) and late (7 D) neutrophils waves with different phenotypical and functional roles, associated with activation or angiogenic activity respectively. Depletion of early infiltrating neutrophils is neuroprotective and has long term positive effects on mouse behaviour, whereas later time point depletion impaired vascular and oligodendrocyte regeneration and correlates with long term anxiety-related behavior. Authors made a very intriguing observation identifying how neutrophil heterogeneity mediates different roles during brain recovery and their potential implication in long term neurocognition. There are some key points that should be reinforced:

Major points:

-Neutrophil depletion model: The authors take advantage of a combination of anti-Ly6G and an Anti-Rat IgG. Author indicate that “we obtained a significant reduction of both, circulating and brain-infiltrated neutrophils, which recovered at day 7 (Suppl. Fig. S2A)” However, they do not show which is the depletion efficiency if depletion takes place at day 7. The authors should check whether the depletion protocol combining anti-Ly6G with anti-rat IgG is efficient also with late depletion. According to the phenotypic profiling, neutrophils from day 7 seems to be more “mature” (e.g. higher expression of CD101) I wonder if this “maturation differences” impacts on the depletion efficiency of the model. In line with this, previous observations on a model of nerve injury shows a neutrophil subpopulation that express relatively low amounts of Ly6G and Ly6B, and relatively high expression of CD14 that were associated with nerve regeneration (Sas et al., Nat Immunol. 2020), authors enriched this population by using anti-CXCR2 treatment.

We thank the reviewer for bringing this point to our attention. We agree that differences in the maturation state and Ly6G expression might affect depletion efficiency. We performed additional experiments and quantified neutrophil numbers in blood and brain at 7, 9 and 13 days after HI, i.e. during and after antibody injection between 6 and 8 days after HI. Compared to depletion at the acute disease phase, we obtained a similar depletion efficiency and recovery (Suppl. Fig S4A-C).

I believe that an alternative model to deplete neutrophils should be included (E.g. iDTR). This will reinforce the central hypothesis of the article and control better for confounding factors such as the neutrophil maturity status or the long-term effect of the antibodies reaching the brain parenchyma (E.g. behavioural tests).

We appreciate the reviewer’s suggestion. However, within the limited time frame of revisions it was not possible to setup this mouse line and to get the required regulatory permissions from the local government to perform these new animal experiments. From our late depletion experiments we do not have indications for a confounding effect by differences in neutrophil maturity. Potential confounding effects by antibodies reaching the brain seem rather unlikely, since sham mice, treated with anti-Ly6G did not show any difference in all of the analyzed readout parameters. Nevertheless, future studies should include the suggested experiments to confirm the present findings. We added this in the discussion section (page 14).

-Moreover, little is known on the effect of the neutrophil depletion model in other leukocytes (e.g. microglial cells).

This is valid concern, In addition to neutrophil counts, we now also quantified the number of CD45^{high}/Ly6G⁻/CD11b⁺ (non-neutrophil myeloid leukocytes) and CD45^{high}/Ly6G⁻/CD11b⁻ (non-myeloid leukocytes), observing no differences in these cell types (Suppl. Fig. S4A,B), suggesting. However, we agree with the reviewer that an intense myeloid-neutrophil crosstalk can modulate disease pathology, as shown in models of adult stroke ^{1,2,3}. Our previous ³ and the present results from Iba-1 immunohistochemistry suggest that neutrophils in the early disease phase promote myeloid cell accumulation/activation (Fig. 3C), though secondary effects due to increased neuronal injury promoted by higher ROS production of early neutrophils cannot be excluded in this *in vivo* setting. Moreover, Iba-1 cannot discriminate between different myeloid cell subsets. Therefore, we characterized myeloid/microglia cell subpopulations 10 days after HI via spectral flow cytometry in more detail. Neonatal HI induced a pronounced switch from resting phenotypes to activated disease-associated macrophage (DAM) phenotypes (Fig. 3D, Suppl. Fig. S6E). Neither early nor late depletion altered these HI-induced responses (Fig. 3D), indicating a limited effect of early and late neutrophil depletion on myeloid cell/microglia diversification in response to neonatal HI.

Neutrophil subsets behaviour:

-In Figure 3F and 3E, the authors state that neutrophils distribution and proximity to vessels differs at 24h and 7d post HI. In figure 3E, they show a clear difference in endothelial cells (CD31 staining). Could the difference in neutrophils localization to vessels be due to a change in endothelial compartment rather than in neutrophils themselves? How the authors envision a possible implication of localization in their model?

We thank the reviewer for this important point. We agree that the increased vessel association of neutrophils at day 7 compared to day 1 may not solely be related to an active response of late neutrophils. This is supported by our new analyses of developmental vascularization, quantifying small and large vessel densities during the period of experimental interventions, i.e. between postnatal day 9 (P9, HI induction) until P16 (7 days after HI). Using light sheet microscopy in large tissue volumes of cleared brain tissues (Suppl. Fig. S7A) followed by automated quantification of vascularization according to our previously established workflow ^{4,5}, we observed that most of the large vessels are established between P9 and P12, while small capillaries develop at later stages, i.e. from P12 onwards (Fig. 4B, Suppl. Fig. S7B). Neonatal HI at P9 induced a strong reduction particularly of larger vessels during the acute disease phase, i.e. 24 h after HI, which, however, recovers until day 7 (Fig. 4A,B, Suppl. Fig. S7A-C). In contrast, development of small vessels remains impaired, as demonstrated by a significantly reduced vessel length density 7 days after HI (Fig. 4B, Suppl. Fig. S7A-C). Nevertheless, compared to day1 overall vessel density is increased in HI animals at day 7, which might indeed explain the closer vessel association at this time point. In addition to an increased capillary density, the structure of the vasculature differs between both time points. One day after HI the basement membrane is closely attached to the endothelial lining, while it is dilated and partially disrupted at day 7 (Fig. 1F, Suppl. Fig. 1D). Basement membrane loosening might provide a unique space keeping neutrophils closer to the vasculature. As such, similar to observations in models of adult stroke ⁶, we frequently observe neutrophils in the perivascular space at day 7 but not at day 1 after HI (Fig. 1F, Suppl. Fig. S1D).

Whether the increased vessel association of late infiltrating neutrophils is causally linked to their pro-angiogenic properties needs to be determined in future studies. However, our new *in vitro* experiments revealed a comparable improvement of endothelial wound healing when incubated with supernatants of neutrophils (Fig. 6J), indicating that direct cell contacts are not needed, which is supported by the fact the Siglec-F^{high} cells were found more frequently in the parenchyma than Siglec-F^{low} cells. We modified our previous hypotheses and conclusions accordingly (page 16).

-The authors identify Siglec-F as a marker for late infiltrating neutrophils but lack any functional further characterization. For example, it would be nice to see whether the Siglec-F^{high} neutrophils have higher angiogenic capacity, their distribution in the damaged brain.

To support our hypothesis, we performed additional *in vitro* experiments with sorted Siglec-F⁺ and Siglec-F⁻ brain infiltrating neutrophils from day 7 after HI. Co-incubation of brain endothelial cells with Siglec-F⁺ neutrophils significantly improved endothelial wound healing compared to their Siglec-F⁻ counterparts (Fig. 6E). Interestingly, Siglec-F⁺ neutrophils showed an increased mRNA expression of angiogenic factors, e.g. VEGF (Fig. 6E) and co-incubation with Siglec-F⁺ cells in the presence of anti-VEGF diminished regenerative effects on endothelial cells (Fig. 6F), suggesting this growth factor being at least partially involved in the regeneration-promoting function of Siglec-F^{high} cells.

Minor:

1. Figure 1B: the image of 24h shows a substantial number of neutrophils in the contralateral hemisphere. This is in apparent contradiction with quantification by imaging and FACS in Supplementary Figure 1A-B. Please change the image if it is not representative. As well, the scale bar is missing.

We exchanged the image with a more representative one. We apologize for the missing scale bar, which is now added in the revised manuscript.

2. Figure 2B: the counterbalance of weight loss in early depleted mice is not clear. It might not be significant but clearly, the trend is the same as in HI mice.

We thank the reviewer for the comment and rephrased the description to “HI induces an acute decrease in weight gain, which was similar in all treatment groups”.

3. Figure 3F: it is not specified at which time point the picture is taken. Show the missing time point to clearly show the difference in localization. Investigate the location and closeness to vessels of Siglec-F positive neutrophils.

We apologize for the missing declaration. We exchanged images and now clearly state the time points (Fig. 1E). Applying multiplex spatial imaging, we now show exemplarily the localization of Siglec-F⁺ neutrophils in relation to different cell types and quantified vessel distances for Siglec-F⁺ and Siglec-F⁻ neutrophils (Fig. 6D).

4. Supplementary Figure 3C: change the scale of the Y axis in the last two graphs to better see the data.

We rescaled the Y-axis as suggested.

Reviewer #2 (Remarks to the Author):

We thank the reviewer for co-reviewing as early career researcher.

Reviewer #3 (Remarks to the Author):

The study examines the role of neutrophils in a neonatal hypoxia-ischemia (HI) model in postnatal day 9 mice, a model of hypoxia-ischemia encephalopathy (HIE) in at-term newborns, by testing effects of neutrophil depletion either early (12/36/50h) or late (6/7/8d) after HI on heterogeneity of peripheral neutrophils and neutrophils in injured brain as well as effects of depletion on early injury in the hippocampus and long-term behavior recovery. The authors demonstrate biphasic recruitment of neutrophils during acute injury (1d) and sub-chronic injury (7 and 9d) after HI and show phenotypical and functional diversity of neutrophils during these injury phases. Importantly, they show that neutrophil depletion early after HI protects neurons in the hippocampus whereas depletion at the later time point promotes death of neurons in the hippocampus, adversely affects oligodendrocyte regeneration and vascular bed, and exacerbates long-term behavior outcome, as manifested by enhanced anxiety-related behavior. The purely injurious role of neutrophils in the pathophysiology of cerebral ischemia in adults and in neonates is being reconsidered. Thus, the significance of this study is in detailing the dynamics of spatio-temporal appearance of neutrophils in injured neonatal brain, demonstration of neutrophil heterogeneity after HI injury and evolved neutrophil phenotypes over time, and in relation to injury. Translational potential of the study is in demonstration that targeting neutrophils during later injury phases, not early when a diagnosis is often yet to be made in newborns, can be beneficial long term. At the same time, there are multiple concerns regarding quality of the data and data presentation, which ultimately affect data interpretation. Conceptually, the gold standard while studying/reporting effects of interventions in cerebral ischemia and hypoxia studies is histological outcome, but there are no data on histological outcomes at any point. Furthermore, presentation of several types of results is questionable and, while the authors acknowledge the dynamic nature of changes in the developing brain, including maturation of the blood-brain barrier interphase and leukocytes, they inappropriately lump the data from contralateral hemisphere (i.e., hemisphere directly affected by hypoxia and remotely by ischemia) obtained at from hours to 10 days after HI. There are also a number of misconceptions regarding cell identities. All together, conclusions are not fully supported by the data and rigor criteria are not met.

In detail:

I. The magnitude of neutrophil accumulation over time. The 3D data on neutrophil presence in injured C57BL/6J pups and some of neutrophil molecules that can affect injury evolution/recovery are novel, meaningful, and enlighten the HI field.

Concerns:

1. The data for contralateral hemisphere are inappropriately lumped for 5 time points, 12h, 24h, 3d, 7d and 10d. Contralateral (hypoxic) hemisphere is known to be affected in the HI model. In fact, the data in Fig1b show sizable neutrophil accumulation in contralateral hemisphere at 24h.

We agree with the reviewer that contralateral hemispheres may not always serve as the appropriate control. To validate our initial results, we performed additional experiments for quantification of neutrophils via light sheet microscopy, now also including sham animals (Fig. 1B). These additional analyses allowed discrimination to “hypoxia-only” effects. For the interest of the reviewer we compared the number of neutrophil numbers in contralateral hemispheres of HI-injured mice to ipsilateral sites of sham-animals (Fig. 1 below).

Fig. 1: No significant neutrophil infiltration in contralateral hemispheres after neonatal HI. Quantification of neutrophils in contralateral hemispheres after HI and ipsilateral hemispheres after sham operations, assessed by whole organ light sheet microscopy in Cubic-cleared brain tissues. n=3-4 sham and n=5-7 HI per time point, two-way ANOVA followed by Šídák's test.

2. Given biological variability in the HI model, n=4 per time point is insufficient. Typically at least n=8-15 pups per group is utilized. Fig. 1B shows vastly different magnitude of neutrophil accumulation in a total of four mice at 7d, from ~2.5x10⁴ cells in one mouse to ~25x10⁴ cells in another mouse and ~9x10⁴ cells in other two mice. Variability is likely related to varying extent of injury in individual mice. Histology data should be presented for 24h and 7d to show variability in the extent of brain swelling at 24h and the magnitude of tissue loss in the cortex and the hippocampus by 7d.

We agree with the reviewer that due to the well-known variability of the injury model, sample sizes need to be appropriate. We performed additional experiments to increase samples sizes, including sham animals (n=19, n=3-4 per time point) and HI mice in these experiments (n=13, n=2-3 per time point) (Fig. 1B). Together with flow cytometry analyses we now included 7-9 sham and 12-18 HI animals per time point for assessment of neutrophil infiltration dynamics with two different techniques, both revealing similar and statistically significant results. Also in view of animal welfare we decided not to increase sample sizes further.

The clearing technique that had to be applied for this kind of analyses, does not allow histological analyses. While tissues cleared with iDisco can be rehydrated followed by conventional immunohistochemistry⁷, this cannot be performed with brain tissues cleared with Cubic. This was needed for imaging of Catchup^{IVM} mice, to retain the signal of tdTomato, which is faded if iDisco clearing is applied. Therefore, conventional histology was technically impossible for these brains. However, to address the question, whether variability in neutrophil infiltration is related to tissue injury, we performed immunohistochemistry analyses to correlate neuronal densities to neutrophil numbers at the two infiltration peaks. While we detected a significant negative correlation at day 1 after HI, no clear association was observed at day 7 (Suppl. Fig. S1A). These new findings support that early neutrophils contribute to acute neurodegeneration while late neutrophils rather promote tissue regeneration, providing an environment for improved long-term neurodevelopment, as reflected by reduced oligodendrocyte and endothelial proliferation at day 10, associated with increased deficits in anxiety-related behavior in the absence of late-infiltrating neutrophils (Fig. 3F, 4D).

3. In Fig. 3D neutrophil numbers in contralateral hemisphere should be added.

According to the first comment of this reviewer, we think sham animals are the relevant control and therefore did not collect contralateral tissues in our previous flow cytometry analyses. Conforming to the 3R principle, we decided not to perform new experiments, only to show contralateral values. Furthermore, neutrophil numbers did not differ between contralateral hemispheres of HI mice and ipsilateral hemispheres of sham mice (Fig. 1 of this document). We now consistently show sham mice for all analyses in the revised manuscript.

4. Fig. 1G. What are the data compared to? If they are compared to the data from shams lumped from all time points, more appropriate analysis should be done after separating the data for different time points.

Values of HI animals were normalized to values of sham animals from the same time point. For clarity, we now included an additional paragraph in the methods section also explaining statistical analysis for these measurements in more detail (page 21). To improve robustness of our initial findings, we increased sample sizes for this assay, adding 2 sham and 1-2 HI mice time point. To increase scientific rigor, we also modified our analyses. While values below local background were initially set to zero and included in statistical analyses, we now excluded them from statistical analyses setting them as n.d. (Fig. 1G,H, source data file). The identification of differentially expressed proteins by ANOVA (comparing normalized values of HI-mice (normalization to sham of the same time point) vs. pooled normalized sham values) included only proteins, for which at least in 2 of 4 sham and 4 of 7-8 HI values per time point were above the detection limit. This analysis and the increased samples size confirmed our initial findings, i.e. biphasic upregulation of selective proteins involved in leukocyte recruitment to sites of tissue injury (e.g. Osteopontin, CCL6) and selective upregulation or recovery of proteins involved in neuroregeneration and angiogenesis at day 7 (e.g. IL-1ra, IL-33, Gas, Amphiregulin, LIF, Angiopoietin-1, Chemerin, VEGF). Moreover, with these more powered and restricted analyses, we also uncovered additional differentially expressed proteins with a selective upregulation at day 7 (Fig. 1G). Among them are neutrophil chemoattractants (e.g. CXCL-2 and GM-CSF) but also regeneration- and angiogenesis-promoting proteins (e.g. IL-13 and TSG-14, Suppl. Table S1).

5. In studying effects on injury, including HI, a vastly different animal number per group, like n=4 in experimental group Vs more than 12-15 in control group, is to be avoided.

We changed data presentation and now compare values of HI mice to values of sham mice of the same time point (Fig. 1A, B).

6. The authors categorically state that neutrophils infiltrate acutely injured HI regions and link this notion to the knowledge in adult stroke models, but it has been documented that neutrophils can signal without entering the parenchyma in stroke and release ROS and other molecules while being adherent to the vasculature. Some studies (Engelhardt) demonstrated that in stroke models the majority of neutrophils accumulate within the neurovascular unit and the subarachnoid space where they remain separated from the brain parenchyma by the glia limitans.

We thank the reviewer for this comment. Taking into account the different infiltration routes of neutrophils and previous studies in adult stroke models, we performed new experiments and in depth analyses of neutrophil localization combining different imaging techniques, including two-photon microscopy (Fig. 1C, Suppl. Fig. S1B) and immunohistochemistry analyses in

tissue sections stained for CD31 (endothelium) / pan Laminin (basement membrane) and Ly6G (Fig. 1F, Suppl. Fig. S1D). The most important finding is that neutrophil localization is strongly dependent on the disease stage. While half of brain neutrophils were found in the vasculature at the acute disease stage, supporting an earlier study in a model of neonatal stroke⁸, up to 80% of neutrophils were located in the parenchyma or transmigrating from the perivascular space (Fig. 1F). Light sheet microscopy data, showing a closer vessel association of late infiltrating neutrophils (Fig. 1E) were confirmed by these analyses, revealing shorter vessel distances of intraparenchymal neutrophils 7 days after HI (Fig. 1F). Assessing regional localization, we found a higher proportion of meningeal neutrophils at day 1 (Fig. 1D). In adult stroke, the meninges specifically the subarachnoid space were reported to be a major localization site / infiltration route^{6,9,10}. Skull-associated and sub-cortical meninges are mostly destroyed during dissection of brain tissues and cannot be assessed via immunohistochemistry. Therefore, we applied tissue clearing to brains with an intact skull followed by two-photon imaging, showing neutrophils in skull-associated meninges in HI but not sham animals (Fig. 1C) We also found a considerable amount of neutrophils in the bone marrow (Suppl. Fig. S1A). Since both localization sites have not yet been reported before in neonatal mice, their relevance needs further attention in future studies, which is beyond the scope of this work. Therefore, we only descriptively address this reviewers' comment. Despite these new observations of meningeal and bone marrow-associated neutrophils, a significant proportion of neutrophils is located in the brain parenchyma, particularly at 7 days after HI (Fig.1E, Suppl. Movie S1). This might not only be related to differences in neutrophil migratory responses but also due to changes in the vascular structure with dilated and partially destroyed basal membranes at 7 days after HI. Delayed basal membrane disruption might explain increased frequency of intraparenchymal neutrophils and basement membrane loosening might provide a unique perivascular space with a specific molecular environment keeping neutrophils closer to the vasculature.

7. Based on appearance of data in Suppl Fig.1, there are 2 subgroups with distinct levels of CCL11, CCL6 and angiopoietins. Was injury pattern/severity same in those mice?

Since we included additional animals and modified data analyses for more robust findings (please refer to our response to comment 4), data point distribution changed. However, injury-related variability is still reflected (Fig. 1G,H; Suppl. Fig. 2B). Therefore, we analyzed brain tissue sections of these animals for apoptosis. While at day 1, except of Chemerin, no clear associations were detected, several significant positive and negative correlations were determined at day 7, the second infiltration peak of neutrophils (Fig. 2 below). These did not only include neutrophil-associated proteins (e.g. MPO) but also typical chemoattractants (e.g. CXCL-1 and CXCL-2) supporting that these signals might have guided neutrophils into the brain at the delayed disease stage. However, among proteins with a significant positive correlation we also detected cytokines with neuroprotective and neuroregenerative functions (e.g. IL-1 α and IL-33, Fig 2 below, Suppl. Table S1), supporting the initiation of endogenous repair mechanisms particularly in severely injured animals. Even though, these are interesting findings, they are rather related to the disease model, without a direct connection to the question to be addressed in the present work. Therefore, and not to distract from the focus this study, we provide this analyses for the interest of the reviewer, only.

Correlation coefficient	d 1	d 3			d 7						
	Chemerin	CCL6	Adipo-nectin	IL-1ra	MPO	CXCL1	CXCL2	CXCL5	CXCL10	IL-1alpha	IL-33
TUNEL+ cells / mm ²	-0.883	0.862	0.867	-0.891	0.715	0.914	0.814	0.928	0.972	0.808	0.727

Fig.2: Correlation between protein abundance and tissue injury at different time points after HI. TUNEL positive cells were quantified via immunohistochemistry and correlated to protein expression data obtained from proteome profiler antibody array analyses (Fig. 1G, H, Supple Fig. S2C). Proteins revealing a significant correlation are summarized in the table and exemplarily plotted.

II. Effects of ‘early’ Vs. ‘late’ neutrophil depletion on long-term functional outcomes. Long-term effects of neutrophils on brain recovery after HI have never been studied. Thus, the data on how neutrophil depletion during individual injury stages affects long-term outcomes are novel and impactful.

Concerns:

1. Data presentation is confusing. For example, Fig. 2C left shows speed in cm/sec in sham and HI mice (average ~6 cm/sec in shams), whereas on the right Y axes is labeled in cm/sec but the data are presented as normalized to 1, which is inconsistent. More importantly, considering highly variable speed in shams, it is more appropriate to show all data as speed and compare values in neutrophil depleted mice to that in shams and/or HI with intact neutrophils.

We agree with the reviewer that all data should be compared in ANOVA analyses and adapted data presentation and statistical analyses accordingly. The main conclusions remained similar.

2. Histology at 5 weeks must be presented and in reference to speed/mobility.

We appreciate the reviewers’ comment, since behavioral outcome does not necessarily correspond to tissue injury. We quantified tissue atrophy in cresyl-violet stained tissue sections, showing a less pronounced HI-induced long-term tissue loss in early-depleted mice, while in late-depleted animals an overall aggravated tissue injury was observed (Fig. 2E). These results are consistent with long-term alteration of HI-induced functional deficits, as demonstrated by a positive correlation with HI-induced altered anxiety-related behavior and hyperactivity, observed in the EPM and OF, respectively (Fig. 2F).

3. One of studied aspects is the effects on oligodendrocyte accumulation, which is important, but the ultimate outcome of oligodendrocyte survival, proliferation and differentiation is development of the white matter. Incorporating effects on the white matter presence/integrity may delineate an important underlying mechanism of recovery and will certainly strengthen the manuscript.

We agree with the reviewer that myelination is key for neurodevelopment. To determine the impact of early and late neutrophil depletion on long-term myelination, we quantified expression of myelin associated proteins via real time PCR and western blot (Fig. 2G,H, Suppl.

Fig. S5E,F). Quantification of mRNA levels for CC1, CNPase, MAG and MBP proteins largely confirmed results from histology and behavioral analyses, demonstrated by a reduced expression in isotype-treated HI mice, which was partially improved by early neutrophil depletion, while late depletion resulted in more severe reductions compared to isotype controls. Reduced oligodendrocyte regeneration detected 10 days after HI might explain long-term myelination deficits and resulting neurological dysfunction in animals with late depletion. Interestingly, differences between sham and isotype-treated animals were less pronounced, indicating that developmental brain maturation partially compensates detrimental effects of neutrophils in the acute disease phase with regard to myelination. However, the impact of late neutrophil depletion remained strong until adolescence, highlighting the importance of neutrophils in the sub-acute disease phase, when regeneration is initiated.

III. Identification of leukocyte subtypes and their roles.

Concerns:

1. Iba-1 cannot discriminate between microglia, freshly accumulated monocytes or differentiated macrophages (Suppl Fig. 3B). Cell type specific markers are to be used to interpret the phenotypes of Iba-1 cells. There is marked accumulation of monocytes that differentiate by 7d in this model.

We thank the reviewer for bringing this point to our attention. We agree that Iba-1 is not exclusively expressed by microglia and that monocytes have been suggested to play an important role¹¹. To determine the impact of early and late neutrophil depletion on myeloid cell populations we performed spectral flow cytometry to distinguish different myeloid/microglia subpopulations. While HI led to the appearance of different disease associated macrophage subsets in the ipsilateral hemisphere 10 days after HI, neither early nor late neutrophil depletion resulted in marked changes of myeloid cell subsets 10 days after HI (Fig. 3D). These data imply a limited impact of neutrophils on these cell types, not excluding a functional interaction between both at other disease time points as suggested in models of adult stroke^{1,2}. This would need further investigations beyond the scope of the present work.

2. CD11b/Ly6C flow cytometry plots (Suppl Fig 4A) are used to gate on neutrophils (Ly6G-high/Ly6C-intermediate Vs. monocytes (Ly6C-high). There are two subpopulations of cells within the chosen box, but the entire Ly6C+ population is chosen, not Ly6C-intermediate. Given such selection and knowing that CD14 is abundantly expressed by monocytes and macrophages is it possible that many of CD14+ cells are in fact monocytes?

We thank the reviewer for this relevant comment, especially since a recent publication described Ly6G⁺ monocytic cells after lung injury¹². To validate that the CD14^{high} cell population in our analysis are in fact neutrophils, we performed two different approaches: We used stricter Ly6C gates followed by gating based on Ly6G and CD115 and analyzed CD14 expression on Ly6C^{high} CD115⁺ Ly6G^{neg} monocytes or Ly6C^{int} CD115⁻ Ly6G⁺ neutrophils showing that CD14 expression follows a similar temporal and disease-dependent regulation in both cell types with the highest levels at 24 h after HI (Fig. 3A below). Additionally, we performed dimensional-reduction by UMAP on a concatenate of both Ly6G⁺ neutrophils and CD115⁺ monocytes. As expected, neutrophils and monocytes clustered in a cell-specific manner. When separating the UMAP by time-point and group and coloring by CD14 expression, high CD14 expression was found exclusively after HI in both neutrophils and Ly6C^{high} monocytes, with higher expression at d1 than at d7 (Fig. 3B, below). Taken together, we are confident that the CD14⁺ population in our analysis are in fact neutrophils as indicated

by their surface phenotype (Ly6G⁺ CD115⁺), which upregulate CD14 in a disease- and time point specific pattern similar to monocytes.

Fig. 3: CD14 expression on neutrophils and monocytes in the brain after neonatal HI. (A) Flow cytometry gating and quantification of Ly6G, Ly6C and CD14 expression on brain neutrophils. (B) UMAP of brain Ly6G⁺ and CD115⁺ cells and projection of Ly6G, CD115, Ly6C and CD44 expression (top). Separation of the UMAP by time point and group and expression of CD14 (bottom).

3. There is substantial interplay between neutrophils and monocytes and/or microglia, interactions that largely mediate injury progression. These interactions need to be considered to support conclusions.

Neutrophil-myeloid cell interactions may play an important role in promotion of ischemic tissue injury as shown in our previous³ and the present work, revealing less Iba-1 immunoreactivity after early neutrophil depletion. Furthermore, our and other recent studies in adult stroke showed that microglia phagocytose neutrophils thereby contributing to resolution of inflammation^{1,2}. However, as mentioned before, characterization of a broad set of myeloid cell activation markers did not show differences by either anti-Ly6G-treatment, which does however not exclude differences in functionality. We added this conclusion when discussing our new data on myeloid cell subsets in the revised manuscript (page 13).

4. The phenotypes of neutrophils upon depletion and reappearance can be very different, which would affect interaction with cells of the monocyte lineage. These mechanisms have not been considered.

Even though we very much appreciate the new hypothesis-generating suggestions, the main focus of this study was on myelination and angiogenesis, not on the interaction between neutrophils and monocytes. We agree that transient anti-Ly6G-mediated depletion may alter the phenotype of reappearing neutrophils. Spectral flow cytometry analyses of recovered neutrophils in the blood showed a slightly more immature phenotype, revealed by reduced expression of CD101, CXCR2 and Ly6G, while a broad set of other markers was, however, not affected (Suppl. Fig. S4D). Keeping in mind the study by Sas et al. reporting an immature neuroprotective neutrophil subset in the setting of adult inflammation-induced optic nerve and

spinal cord injury¹³, the reappearing immature neutrophil subset might be responsible for the protective effect after early depletion. However, newly appearing neutrophils do not show this neuroprotective phenotype, i.e. elevated CD14 expression. Furthermore, our previous findings by Mülling et al. showed a strong neuroprotective effect of acute neutrophil depletion, assessed already 2 days after HI before reappearance of neutrophils. Though being slightly different, reappearing neutrophils seem to acquire similar regenerative functions as in isotype-treated animals, e.g. promoting oligodendrocyte and endothelial proliferation and improving long-term myelination. The detrimental effect of early neutrophils in the acute disease phase, however, also remains obvious, as shown by an improved functional outcome in the open field.

IV. Neutrophil –mediated protection against sub-chronic injury and effects on angiogenesis.

Overall the data clearly demonstrate protection of hippocampal neurons by the presence of neutrophils (Fig. 3B) and increased oligodendrocyte maturation (CC1+) 10d after HI.

Concerns:

1. Data presentation in this figure suffers the same problem as in Fig. 2, i.e., the number of neurons is highly variable in individual mice with intact neutrophils, thus, expressing the data in mice with depleted neutrophils relative to “1” is uninformative and can be misleading.

We agree with the reviewer that all data should be compared in one ANOVA analyses and adapted data presentation and statistical analyses accordingly.

2. Based on appearance of Olig2+/CCL1+ and Olig2+/Ki67+ data in Fig. 3C&D there are 2 subgroups. Was injury volume/distribution similar in individual mice in those subgroups?

To determine, whether different responses in mature and proliferating oligodendrocytes are associated with a different overall tissue injury, we performed correlation analyses between these readouts and tissue as well as neuronal loss. This is an important and relevant question, but similarly as for comment 7 in the first part, it is rather related to the disease model, not directly linked to the focus of the present study. Therefore, we provide these analyses for the interest of the reviewer in Fig. 4 below. Including all experimental groups, we indeed detected significant positive correlations between the number of CC1+ oligodendrocytes and neurons, CC1+ cells and the hemisphere tissue area, Ki67+ oligodendrocytes and TUNEL+ cells (Fig. 4A below). These results suggest that HI-induced oligodendrocyte cell loss goes along with neuronal loss and that more pronounced regenerative oligodendrocyte proliferation responses are induced in more severely affected animals. However, correlation coefficients were rather small between 0.31 and 0.53. Therefore, these associations may not solely explain variability. This is supported by the fact, that the two “subgroups” the reviewer is referring do not show a clear relation between the different outcome parameters (Fig. 4C,D below). For instance, out of the 5 animals with high CC1+ cell number only two showed high NeuN numbers, while two revealed lower neuronal densities below median values of the total group (Fig. 4C). Similarly, the 4 animals with high Ki67/Olig2+ cell numbers showed a variable degree in the number of TUNEL+ ranging between highest and smallest values (Fig. 4D). These data implicate that more complex mechanisms, most likely also involving other cell types contribute to variability in this injury model.

Fig. 4: Correlation analysis of oligodendrocytes with tissue injury. (A/B) Spearman correlation analyses between Olig2/CC1⁺ or Olig2/Ki67⁺ cells and NeuN⁺ cells, TUNEL⁺ cells and hemisphere tissue area. n=66. (C/D) Relationship of Olig2/CC1⁺ cells or Olig2/Ki67⁺ cells with NeuN⁺ or TUNEL⁺ cells in early depleted HI mice. Mice with high in Olig2/CC1 cell numbers are highlighted in blue (C), mice with high Olig2/Ki67 cell number are highlighted in green (D). n=10.

3. The results related to vessel coverage are unclear. What was measured in Fig.3E, pixel number? Vessel size distribution, including establishment of capillary networks that occurring during a studied time frame should be reported. Was there measurable number of tip cells?

We thank the reviewer for this important point and apologize for unclear description of our initial immunohistochemistry analyses for CD31 quantification, which was indeed coverage, i.e. % positive area. We agree that postnatal developmental vascularization needs to be considered in interpretation of the timed interventions. To overcome limitations of 2D immunohistochemistry analyses and to provide detailed information on vessel sizes, we performed new experiments to quantify vascularization in larger tissue volumes via light sheet microscopy based on previous studies^{2,4}. These analyses revealed that total vessel length density is continuously increasing with development, which is disturbed by neonatal HI at P9 (Fig. 4 A,B, Suppl. Fig. S7A-C). Vessel branching is particularly increased between P12 and P16 in sham mice, but not in HI-injured mice (Suppl. Fig. S7B). To gain deeper insight into dynamics of small capillary and larger caliber vessel development, we quantified vessel length densities of vessels smaller and larger than 7 μm based on Wälchli et al.¹⁴. Most of the large vessels are established between postnatal day 9 and 12, while small capillaries develop at later stages, i.e. from postnatal day 12 onwards (Fig. 4B, Suppl. Fig. S7A-C). Neonatal HI at P9 induced a strong reduction particularly of larger vessels at the acute disease phase, i.e. 24 h after HI, which, however, recover until day 7 (Fig. 4B, Suppl. Fig. S7A-C). In contrast, development of small vessels remains impaired, demonstrated by a significantly reduced vessel length density 7 days after HI (Fig. 4B, Suppl. Fig. S7A-C). Delayed capillary regrowth might be related to an early impact on Tip cell numbers. Quantification of Tip cell numbers

showed a unique developmental regulation in sham mice, with a considerable number of cells being present between P9 and P10, but rapidly declining from P12 until P16, when no cells could be detected anymore (Suppl. Fig. S7D). Neonatal HI induces an acute Tip cells loss within the first 24 hours after insult, which recovers until 3 days after HI (Suppl. Fig. S7D).

The impact of neutrophils on HI-induced vascularization deficits was investigated 10 days after HI. (Fig. 4 C). Compared to day 7 after HI, small vessel development slightly recovered in isotype-treated HI mice, which was, however, strongly impaired in the absence of late infiltrating neutrophils (Fig. 4C). Since Tip cells recovered by 3 days after HI (Suppl. Fig. S6D) these cells seem rather unlikely to be targeted by late infiltrating neutrophils. However, in addition to vascular sprouting from Tip cells, a major mechanism of post-ischemic vascular remodeling is the generation of new endothelial cells. Indeed, analyses of endothelial cell proliferation via immunohistochemistry demonstrated an endogenous increase in Ki67+ endothelial cells, which was, however, reduced in animals with late neutrophil depletion (Fig. 4D). Together, these data suggest, that early-infiltrating neutrophils contribute to acute HI-induced large vessel loss, while late-infiltrating neutrophils promote vascular regeneration, especially of small capillaries.

V. The data in aortic ring assay are elegant but there are not really relevant to the studied effects in the brain as neural vasculature is so distinct in many regards including presence of the BBB.

We agree with the reviewer that aortic rings are limited in reflecting responses of brain endothelial cells. Therefore, we performed all further experiments with the brain endothelial cell line bEnd.3, quantifying wound healing responses, i.e. endothelial migration in scratch assays (Fig. 5D, Fig. 6E,J).

VI. Sex differences are not considered.

The present study was not intended to determine sex differences and was, therefore, not powered enough. According to the Journal's policy "*Authors should refrain from conducting post hoc sex- and gender-based analysis if the study design is insufficient (for example, low sample size) to enable meaningful conclusions.*", we present sex-stratified analyses only for the interest of the reviewer in Fig. 5 below, focusing on readouts with the largest samples sizes, i.e. immunohistochemistry data from 10 days after HI and long-term outcome data. Overall we observed similar regulations in female and male mice. The effect of late neutrophil depletion on endothelial cell and oligodendrocyte proliferation as well as long-term myelination was slightly stronger in males. Females revealed more severe long-term tissue atrophy after late neutrophil depletion. The impact of HI on long-term behavioral alterations seems less pronounced in males, which can be however attributed to higher variability in male sham animals, supporting that larger samples sizes will be needed to properly address the question of sex differences.

Fig. 5: Sex-stratified analyses of immunohistochemistry and neurobehavioral outcome data. Immunohistochemistry analyses of neuronal density, Iba-1, oligodendrocyte and endothelial proliferation were assessed 10 days after HI. Tissue atrophy analyzed in cresyl violet-stained tissue sections and myelin basic protein expression, assessed via western blot was analyzed 6 weeks after HI following behavioral testing in the Elevated Plus Maze (anxiety) and the Open Field test (hyperactivity). One-way ANOVA followed by Šídák's multiple comparisons test.

Reviewer #4 (Remarks to the Author):

The manuscript by Ritcher et al and entitled: Neonatal Hypoxic-ischemic brain injury sequentially recruits neutrophils with dichotomous phenotype and function, is a fine study characterising the biphasic influx of neutrophils with distinct characteristics in neonate brains post HI. This is a very informative study that provide some novelties: specifically, the authors' data suggest that early neutrophil influx is detrimental whilst late neutrophil influx is beneficial (associated with angiogenesis), responses associated with their phenotypical characteristics and the selective depletion. There are however outstanding questions that the authors should consider:

The authors should provide more details on the mechanisms of recruitment of neutrophils at the different timepoints. Whilst this study gives some insights into the indirect mediators present in the brain during neutrophil recruitment, they have not investigated the classical neutrophils chemoattractants at the two time points of the response (i.e. CXCK1/2, Lix, CXCL12 etc, LTP4, PAF, FPR agonists etc..). Of note, Zhang et al (Front. Neuro 2021) demonstrated the role of GM-CSF in the early recruitment of neutrophils, the current authors have not discussed or investigated this possibility either; and in the light of their different phenotypic characteristics, the authors should provide stronger mechanistic insight into the recruitments of the neutrophils rather than classical inflammatory mediators.

We thank the reviewer for this comment. Reanalysis and increasing samples sizes of proteome profiler antibody arrays (please also refer to our response to comment I-4 of reviewer 3) led to the identification of further differentially expressed proteins at the different time points (Fig. 1G,H). These also include classical neutrophil chemoattractants, like CXCL-1, CXCL-2, LIX and, interestingly, also GM-CSF (Fig. 1G, Fig. 6G). The unique upregulation of CXCL-1 at 24 hours and downregulation at 7 days may explain early neutrophil infiltration, while the selective induction of CXCL-2, LIX and GM-CSF at 7 days may have contributed to the second neutrophil infiltration wave. However, we agree that in addition to signals from the brain, neutrophil responses itself may differ between the studied time points due to maturation. Therefore, we made use of our recently developed multi-lens array microscope ComplexEye designed for high-throughput and high-resolution imaging of cell migration ¹⁵. We screened neutrophil migration responses to a variety of migratory factors, including the ones suggested by this reviewer (Fig. 1J,K). These results together with proteome profiler data led to interesting findings. For instance, while LIX and GM-CSF are upregulated in brain tissues 7 days after HI (Fig. 1G,H; Fig 6G, Suppl. Fig. S2B), neutrophil responses to LIX and GM-CSF stimulation were more pronounced in neutrophils isolated 1 day after HI (Fig. 1J,K, Suppl. Fig. S3). In contrast, cerebral CXCL-1 is upregulated at day 1 (Fig. 1H), while neutrophil responses to CXCL-1 stimulation were stronger at day 7 (Fig. 1J,K). For CXCL-2 we observed a biphasic response to stimulation (Fig. 1J,K), while this chemokine is selectively upregulated in the brain at 7 days after HI (Fig. 1G). Together, these findings suggest that both, the different tissue environment and developmental differences in neutrophil responses to chemoattractant stimulation, contributed to the biphasic infiltration pattern of neutrophils.

- Whilst the authors showed a clear influx of neutrophils in the brain at day 24 and D7, it would be informative if they could give more details on the location of those neutrophils using higher resolution images: are they solely peri- or intravascular? Do they locate in a specific structure of the brain, same for both influx or in different locations? The images provided are unclear. Of note, the study by Shrivastava et al (neuro Res Int 2012, not cited) demonstrated already the

predominance of neutrophils in the neonatal brain in specific regions at day-7 that is different from the neutrophil influx at the early time-point post HI.

We appreciate the reviewer's comment and performed in depth characterization of neutrophil localization using different 3D and 2D imaging techniques (Fig. 1C-F, Suppl. Fig. S1). Two-photon imaging in brain tissues with intact skull showed neutrophils in the subcortical meninges and skull bone marrow (Fig. 1C dorsal view, Suppl. Fig. S1B). However, this imaging method also confirmed a significant intraparenchymal tissue infiltration at both time points (Fig. 1C). Confocal microscopy in 20 μm tissue sections stained for CD31, Laminin and anti-Ly6G enabled a more in depth analysis, distinguishing intravascular, perivascular, transmigrating and clearly intraparenchymal neutrophils (Suppl. Fig. S1D, Fig. 1F). At 1 day after HI, more than half of neutrophils were located in the vasculature, while up to 60% were localized in the parenchyma and up to 20%, either within or crossing the barrier of the perivascular space at 7 days after HI (Fig. 1F, Suppl. Fig. 1D). Confirming results from 3D analyses (Fig. 1E), we detected smaller vessel distances of intraparenchymal neutrophils at day 7 (Fig. 1F).

Characterization of regional neutrophil distribution showed a disease-stage-dependent accumulation in certain brain areas with a high frequency in the thalamus and the meninges between the hippocampus and thalamus at the acute disease stage, while almost half all neutrophils were detected in the hippocampus at 7 days (Fig. 1F, Suppl. Fig. 1SC), suggesting the inner meninges as a relevant infiltration route to both, the hippocampus and the thalamus. Even though these data partially confirm the work by Shrivastava et al ¹⁶, showing a strong infiltration in the hippocampus one week after HI, time-dependent regulations in the thalamus seem to differ. In the present work, we observed more neutrophils in the thalamus at day 1 compared to day 7. About the reasons we can only speculate, but differences in age (P7 vs. P9 in the present work) and experimental models (8% vs. 10% O₂ in this study) may account for these differences.

- What are the functional characteristics of late-infiltrated neutrophils? the authors provided details on their phenotypical characteristics and the effect of their depletion, but one could argue that the tissue vascularisation response for the later in an indirect phenomenon. For instance, are the pro-angiogenic neutrophils responsible for the high VEGF detected at 7 days (if so, they could consider using VEGF conditional KO)? Or is it through a different molecular mechanism (e.g. IL-4)? Depletion is insufficient to demonstrate the functionality of those cells (as this could be an indirect response due to the lack of their presence or death in the tissue etc..).

We thank the reviewer for this important point. We agree that cause and consequence effects cannot be entirely clarified *in vivo*. Therefore, we performed a variety of new *in vitro* functional assays co-incubating brain endothelial cells with neutrophils. While co-culture with brain neutrophils from day 1 did not modulate spontaneous wound healing (i.e. migration), 7d-neutrophils resulted in an increase by 50% compared to untreated cells (Fig.5D). Based on the unique emergence of Siglec-F^{high} neutrophils in late-infiltrating neutrophils, we compared both cell populations in more detail (Fig. 6). Interestingly, Siglec-F⁺ neutrophils showed an increased mRNA expression of angiogenic factors, e.g. VEGF (Fig. 6E) and co-incubation with Siglec-F⁺ cells in the presence of anti-VEGF diminished regenerative effects on endothelial cells (Fig. 6F), suggesting this growth factor being at least partially involved in the regeneration-promoting function of Siglec-F^{high} cells. Nevertheless, further studies will be needed to verify the functional relevance of the identified neutrophil subset *in vivo*. This would require time-and

cell-type-specific knockout animals, which at least to our knowledge, are not commercially available until now.

- Regarding the effect of the neutrophil depletion on neurons, are there changes in neuronal cells due to lack of proliferation or apoptosis?

To determine whether the slightly increased neuronal loss in late-depleted animals 10 days after HI was associated with secondary apoptosis or neurogenesis, we performed new immunohistochemistry analyses (Suppl. Fig. S6A-C). The number of TUNEL positive cells was elevated in HI-injured animals at this delayed time point. However, during this delayed disease phase the extent of apoptosis is very low compared to early time points, as demonstrated by a 10times smaller number of TUNEL⁺ cells in the present study (10 days after HI) compared to our previous work (2 days after HI, Mülling et al.) (Suppl. Fig. S6A). Anti-Ly6G treatment did not modulate these responses. With regard to neurogenesis, the number of proliferating cells in the neurogenic subgranular zone and the number of doublecortin positive cells was elevated in all HI groups independent of early or late neutrophil depletion (Suppl. Fig. S6B,C). These results suggest, that increased neuronal numbers in mice with early neutrophil depletion resulted from an improved survival of neurons that didn't die in the acute disease phase. Furthermore, neurons and/or neuronal precursor cells are not a primary target of neutrophils at the delayed disease stage. Impaired endothelial cell and oligodendrocyte proliferation after late depletion rather suggest that late-infiltrating neutrophils provide an environment for regeneration supporting ongoing developmental processes, preventing functional neurodevelopmental deficits, i.e. alteration of anxiety-related behavior.

We are confident that if the authors answer those simple questions, the manuscript will be greatly strengthened.

We thank the reviewer for the encouraging comment.

--

References

1. Neumann J, *et al.* Very-late-antigen-4 (VLA-4)-mediated brain invasion by neutrophils leads to interactions with microglia, increased ischemic injury and impaired behavior in experimental stroke. *Acta Neuropathol* **129**, 259-277 (2015).
2. Otxoa-de-Amezaga A, *et al.* Microglial cell loss after ischemic stroke favors brain neutrophil accumulation. *Acta Neuropathol* **137**, 321-341 (2019).
3. Mulling K, *et al.* Neutrophil dynamics, plasticity and function in acute neurodegeneration following neonatal hypoxia-ischemia. *Brain Behav Immun* **92**, 232-242 (2021).
4. Hagemann N, *et al.* Microvascular Network Remodeling in the Ischemic Mouse Brain Defined by Light Sheet Microscopy. *Arterioscler Thromb Vasc Biol* **44**, 915-929 (2024).
5. Spangenberg P, *et al.* Rapid and fully automated blood vasculature analysis in 3D light-sheet image volumes of different organs. *Cell Rep Methods* **3**, 100436 (2023).
6. Enzmann G, *et al.* The neurovascular unit as a selective barrier to polymorphonuclear granulocyte (PMN) infiltration into the brain after ischemic injury. *Acta Neuropathol* **125**, 395-412 (2013).
7. Mohamud Yusuf A, *et al.* Light Sheet Microscopy Using FITC-Albumin Followed by Immunohistochemistry of the Same Rehydrated Brains Reveals Ischemic Brain Injury and Early Microvascular Remodeling. *Front Cell Neurosci* **14**, 625513 (2020).
8. Fernandez-Lopez D, *et al.* Blood-brain barrier permeability is increased after acute adult stroke but not neonatal stroke in the rat. *J Neurosci* **32**, 9588-9600 (2012).
9. Marchetti L, Engelhardt B. Immune cell trafficking across the blood-brain barrier in the absence and presence of neuroinflammation. *Vasc Biol* **2**, H1-H18 (2020).
10. Planas AM. Role of Immune Cells Migrating to the Ischemic Brain. *Stroke* **49**, 2261-2267 (2018).
11. Mallard C, Ferriero DM, Vexler ZS. Immune-Neurovascular Interactions in Experimental Perinatal and Childhood Arterial Ischemic Stroke. *Stroke* **55**, 506-518 (2024).

12. Ruscitti C, et al. Recruited atypical Ly6G(+) macrophages license alveolar regeneration after lung injury. *Sci Immunol* **9**, eado1227 (2024).
13. Sas AR, et al. A new neutrophil subset promotes CNS neuron survival and axon regeneration. *Nat Immunol* **21**, 1496-1505 (2020).
14. Walchli T, et al. Hierarchical imaging and computational analysis of three-dimensional vascular network architecture in the entire postnatal and adult mouse brain. *Nat Protoc* **16**, 4564-4610 (2021).
15. Cibir Z, et al. ComplexEye: a multi-lens array microscope for high-throughput embedded immune cell migration analysis. *Nat Commun* **14**, 8103 (2023).
16. Shrivastava K, Chertoff M, Llovera G, Recasens M, Acarin L. Short and long-term analysis and comparison of neurodegeneration and inflammatory cell response in the ipsilateral and contralateral hemisphere of the neonatal mouse brain after hypoxia/ischemia. *Neurol Res Int* **2012**, 781512 (2012).

Reviewer #1:

Authors have addressed the main concerns raised in my revision and additional experiments included reinforce the conclusions of this study.

We thank the reviewer for re-evaluation and the positive feedback on our revised manuscript.

Reviewer #2:

We thank the reviewer for repeated co-reviewing as early career researcher.

Reviewer #4:

I thank the authors for providing clear answers to my comments and adding the new results to their original manuscript.

We thank the reviewer for re-evaluation and the positive feedback on our revised manuscript.

Reviewer #5:

Thank you for this excellent manuscript and very comprehensive well-written overview of this important area.

Minor questions: Can you mention interspecies differences in neutrophil counts and function if relevant to explain the gap in knowledge to date and the further relevance of your research? In view of the fact that in mice only 20% of leukocytes are neutrophils and this is 80% in humans this may imply that your research may be even more relevant in human infants. It would be helpful to link with some of the human data on VEGF, GFAP and GM-CSF in the relevant patient group.

We thank the reviewer for the very positive feedback and appreciate the suggestion to include a short discussion regarding clinical translation. We agree that interspecies differences have to be taken into account, as recently shown in comparative transcriptomic and proteomic analyses between adult human and murine neutrophils (Hackert et al. 2023, Ghosh et al. 2024). Despite these differences, we observed a considerable overlap of 81% proteins expressed in both species; the majority of orthologous proteins involved in immune response pathways (Ghosh et al. 2024). Considering a larger proportion of neutrophils in humans specifically in newborns compared to rodents and adults, it will be important to follow up our experimental findings in the clinical setting of neonatal encephalopathy (NE). This is supported by observational studies in newborns with NE, revealing that disease-stage specific decreases in serum VEGF and GM-CSF levels were associated with abnormal MRI and worse outcomes (Chavez-Valdez et al. 2021, Sweetman et al. 2020). Though supporting our findings, these data were derived from blood samples and analyses time points can hardly be translated to rodents. Further longitudinal clinical studies with samples from the cerebrospinal fluid combined with comprehensive blood parameter analyses in rodent HI-models might bridge the gap to clinical translation. We included these thoughts in the revised discussion section (line 538-549).

References:

Chavez-Valdez R, et al. Therapeutic Hypothermia Modulates the Relationships Between Indicators of Severity of Neonatal Hypoxic Ischemic Encephalopathy and Serum Biomarkers. *Front Neurol* 12, 748150 (2021).

Ghosh S, et al. Proteomic Characterization of 1000 Human and Murine Neutrophils Freshly Isolated From Blood and Sites of Sterile Inflammation. *Mol Cell Proteomics* 23, 100858 (2024).

Hackert NS, et al. Human and mouse neutrophils share core transcriptional programs in both homeostatic and inflamed contexts. *Nat Commun* 14, 8133 (2023).

Sweetman DU, et al. Neonatal Encephalopathy Is Associated With Altered IL-8 and GM-CSF Which Correlates With Outcomes. *Front Pediatr* 8, 556216 (2020).